# Aortic pathology from protein kinase G activation is prevented by an antioxidant vitamin B$_{12}$ analog

Gerburg K. Schwaerzer[1,4], Hema Kalyanaraman[1,4], Darren E. Casteel[1], Nancy D. Dalton[1], Yusu Gu[1], Seunghoe Lee[1], Shunhui Zhuang[1], Nisreen Wahwah[1], Jan M. Schilling[2], Hemal H. Patel[2], Qian Zhang[1], Ayako Makino[1], Dianna M. Milewicz[3], Kirk L. Peterson[1], Gerry R. Boss[1] & Renate B. Pilz [1]

People heterozygous for an activating mutation in protein kinase G1 (*PRKG1*, p.Arg177Gln) develop thoracic aortic aneurysms and dissections (TAAD) as young adults. Here we report that mice heterozygous for the mutation have a three-fold increase in basal protein kinase G (PKG) activity, and develop age-dependent aortic dilation. *Prkg1*$^{R177Q/+}$ aortas show increased smooth muscle cell apoptosis, elastin fiber breaks, and oxidative stress compared to aortas from wild type littermates. Transverse aortic constriction (TAC)—to increase wall stress in the ascending aorta—induces severe aortic pathology and mortality from aortic rupture in young mutant mice. The free radical-neutralizing vitamin B$_{12}$-analog cobinamide completely prevents age-related aortic wall degeneration, and the unrelated anti-oxidant N-acetylcysteine ameliorates TAC-induced pathology. Thus, increased basal PKG activity induces oxidative stress in the aorta, raising concern about the widespread clinical use of PKG-activating drugs. Cobinamide could be a treatment for aortic aneurysms where oxidative stress contributes to the disease, including Marfan syndrome.

[1] Department of Medicine, University of California San Diego, La Jolla, CA 92093, USA. [2] Department of Anesthesiology, University of California San Diego, La Jolla, CA 92093, USA. [3] Division of Medical Genetics and Cardiology, Department of Internal Medicine, University of Texas Health Science Center at Houston, Houston, TX 77030, USA. [4]These authors contributed equally: Gerburg K. Schwaerzer, Hema Kalyanaraman. Correspondence and requests for materials should be addressed to R.B.P. (email: rpilz@ucsd.edu)

Aortic aneurysms account for 1–2% of deaths in Western countries, and despite improvements in surgical repair, morbidity and mortality remain high, especially with thoracic aortic aneurysms and dissections (TAAD)[1,2]. Blood pressure control with β-adrenergic or angiotensin receptor blockers modestly improves prognosis[1–3]. The thoracic aortic pathology in TAAD is characterized by progressive elastin fiber fragmentation, smooth muscle cell (SMC) loss, and collagen accumulation, leading to aortic dilation and/or rupture[2]. Mutations causing hereditary TAAD affect proteins regulating transforming growth factor-β (TGF-β) signaling, e.g., TGF-β receptors-1 and -2 in Loeys–Dietz syndrome and fibrillin-1 in Marfan syndrome, or components of the SMC contractile apparatus, e.g., SM-α-actin (ACTA2) and myosin heavy chain-11 (MYH11)[2,4]. Aortic pathology has been attributed to SMC de-differentiation and activation of stress pathways leading to increased production of tissue-destructive matrix metalloproteinases (MMPs)[2,5–9].

PKG1 mediates vasodilation in response to nitric oxide (NO)-induced cGMP synthesis[10,11]. It also regulates SMC differentiation, preventing a de-differentiated, proliferative phenotype after vascular injury[12–15]. Excess NO may contribute to aortic disease in Marfan syndrome and other forms of TAAD, but it is unknown whether NO's effects are mediated by PKG[16]. Case reports suggest that long-term use of PKG-activating drugs—e.g., phosphodiesterase-5 inhibitors for erectile dysfunction or pulmonary hypertension—may increase risk for acute aortic dissections[17–21].

We previously identified a recurrent gain-of-function mutation in PRKG1 as a cause of early-onset thoracic aortic disease in humans[22]. Here we show that mice carrying this mutation recapitulate the human disease, and that pathological changes in the aorta are, at least in part, caused by oxidative stress, since they are prevented by treating the mice with two structurally- and mechanistically- unrelated anti-oxidants.

## Results

### PKG activity and blood pressure in $Prkg1^{R177Q/+}$ mice.

The PRKG1,p.Arg177Gln mutation associated with familial TAAD is located in the first cGMP-binding domain of the kinase and causes constitutive activation of both PKG1α and 1β isoforms (Supplementary Fig. 1a–f). The catalytic domain is unchanged, and the mutant protein (hereafter referred to as PKG1$^{RQ}$) is still inhibited by the PKG-specific peptide DT2 (Supplementary Fig. 1g)[23]. About 50% of homozygous mice (referred to as $Prgk1^{RQ/RQ}$) died within the first 6 weeks of life from gastrointestinal dysfunction, with dilated esophagus, stomach, and bowel. Surviving $Prkg1^{RQ/RQ}$ mice were underweight and smaller compared to wild-type (WT) littermates, even when given liquid food after weaning (Supplementary Fig. 1h, i). We focused on heterozygous $Prkg1^{RQ/+}$ mice, because heterozygous humans manifest disease[22]. $Prkg1^{RQ/+}$ mice appeared healthy, weighed the same as WT littermates, and had no obvious developmental abnormalities of the aorta (Supplementary Fig. 1h–l).

PKG1 protein and cGMP-stimulated PKG activity were the same in aortas from $Prkg1^{RQ/+}$ mice and their WT littermates, but basal PKG activity in the absence of cGMP was about three-fold higher in aortas from mutant mice, when measured with a peptide substrate (Fig. 1a) or by assessing phosphorylation of vasodilator-stimulated phosphoprotein (pVASP; Fig. 1b). Although the purified mutant enzyme is fully active in the absence of cGMP (Supplementary Fig. 1f), basal PKG activity in heterozygous aortas was <50% of total cGMP-stimulated activity (Fig. 1a), likely because the wild-type enzyme inhibits the mutant enzyme in a heterodimer[22]. Similar to the purified enzyme, basal and cGMP-stimulated PKG activities in the aorta were largely inhibited by DT2 (Fig. 1a).

Humans carrying the $PRKG1^{R177Q}$ mutation are generally normotensive[22], and consistent with this finding, $Prkg1^{RQ/+}$ mice had only modestly reduced systolic and diastolic blood pressures compared to wild-type littermates, resulting in ~10 mm Hg lower mean arterial pressures during sleep and wake hours (Fig. 1c, d, Supplementary Fig. 2a). Heart rate and pulse pressure were not altered (Supplementary Fig. 2b, c). Thus, although acute PKG1 activation by NO/cGMP causes vasodilation and hypotension in mice and humans[11], a sustained increase in basal PKG1 activity has a small effect on blood pressure.

We evaluated the physiologic consequences of the PKG1$^{RQ}$ mutant protein by assessing contraction of aortic rings derived from mutant and wild-type mice. The concentration of prostaglandin F2α required to pre-contract rings was similar in aortas from eight month-old-wild type and mutant mice (2.15 ± 0.17 versus 2.27 ± 0.86 μM; means ± SEM, $n = 5$ wild type and mutant vessels, respectively). However, relaxation of the rings in response to 8-CPT-cGMP or to acetylcholine—which is largely NO-mediated[24]—was reduced in the mutant vessels compared to wild type vessels (Supplementary Fig. 2d, e). These data are consistent with higher basal, cGMP-independent PKG activity in the aortas of heterozygous mice compared to wild-type mice, resulting in smaller cGMP-induced increases in enzyme activity (Supplementary Fig. 2f). Reduced NO bioavailability due to oxidative stress in the $Prkg1^{RQ/+}$ mice (described below) may contribute to decreased acetylcholine-induced aortic relaxation, but this requires further investigation.

### Age-dependent pathologic changes in $Prkg1^{RQ/+}$ aortas.

The $Prkg1^{RQ/+}$ mice had normal thoracic aorta dimensions at 4–6 months of age, but moderate aortic dilation became apparent at 12 months, with similar phenotypic changes in males and females (Fig. 1e, f, Supplementary Fig. 2g–i). While the aortas of young $Prkg1^{RQ/+}$ mice were histologically normal, the aortas of 12-month-old mutant mice had increased elastin fiber breaks, SMC apoptosis and loss of SMCs, and collagen accumulation in the media layer; media thickness was similar as in WT mice (Fig. 1g–j, Supplementary Fig. 2j). Thus, middle-aged $Prkg1^{RQ/+}$ mice recapitulated aortic changes observed in patients heterozygous for the PKG$^{RQ}$ mutation[22,25].

### Gene expression, oxidative stress, and MMP activity.

Compared to WT littermates, aortas from 4-month-old $Prkg1^{RQ/+}$ mice showed increased mRNAs encoding the SMC contractile proteins Acta2, Myh11, and Tagln, and the transcription factor myocardin, consistent with previous in vitro findings that PKG1 promotes a differentiated, contractile SMC phenotype (Fig. 2a)[12–14]. Expression of TGF-β1 and its target genes Ctgf, Serpine1, Eln, and Nox4 were also increased in the mutant mice, while Col3a1 showed a modest non-significant increase; expression of the proteoglycans lumican (Lum) and decorin (Dcn) was unchanged (Fig. 2a, b). Increased TGF-β signaling occurs in other heritable diseases with TAAD, including Marfan syndrome[9,26,27]. However, the basis of increased TGF-β signaling, and whether it has a causative or compensatory role remains a matter of debate[2,3,5,9,28].

NADPH oxidase-4 (Nox4) mRNA was ~10-fold higher in aortas from $Prkg1^{RQ/+}$ compared to WT mice, but Nox2 was not altered and Nox1 was below detection (Fig. 2b). Nox4 is regulated primarily at the transcriptional level; the enzyme is constitutively active and generates mainly hydrogen peroxide ($H_2O_2$), with some superoxide ($O_2^-$)[29]. SMCs isolated from $Prkg1^{RQ/+}$ aortas produced more $H_2O_2$ than those from WT aortas, and aortas

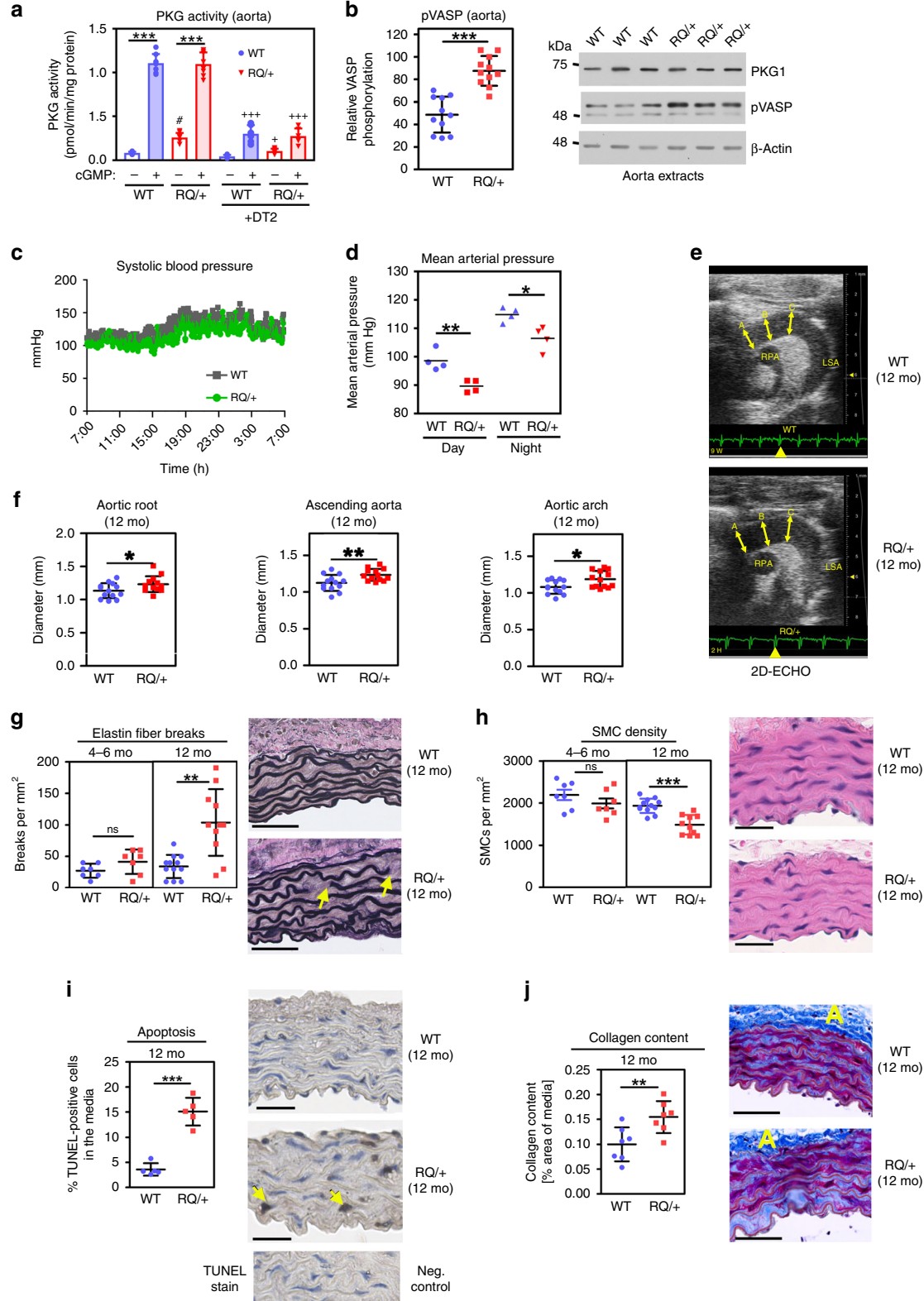

from *Prkg1*[RQ/+] mice showed increased oxidative stress, evident by increased dihydroethidium staining and DNA-, lipid-, and protein-oxidation products; *Prkg1*[RQ/+] mice additionally had higher serum ascorbyl radical concentrations than WT mice (Fig. 2c–g, Supplementary Fig. 3a, b). c-Jun N-terminal kinase (JNK) appears important in the pathogenesis of TAAD, because

JNK upregulates MMPs and induces SMC apoptosis, and JNK inhibitors reduce aortic aneurysms in Marfan mice and in other mouse models of TAAD[7,30–32]. We found increased activity of both JNK and MMPs in *Prkg1*[RQ/+] compared to WT aortas (Fig. 2h–j), consistent with JNK activation by $H_2O_2$ and via MEKK1 activation by PKG1 (refs. [30,32–34]).

**Fig. 1** Development of age-related pathologic changes in the thoracic aortas of $Prkg1^{RQ/+}$ mice. **a, b** PKG activity was measured in aortic extracts from wild type (WT, blue symbols) and $Prkg1^{RQ/+}$ (RQ/+, red symbols) mice using a synthetic peptide (**a** $n = 2$ M + 6 F mice per genotype) or by assessing phosphorylation of vasodilator-stimulated protein (VASP) using a phospho-Ser$^{239}$-specific antibody (**b** $n = 5$ M + 6 F mice per genotype). The PKG inhibitor DT was added as indicated. Three wild type and three mutant aortas are shown in the western blot. **c, d** Systolic blood pressure measured by telemetry over 24 h in 4-month-old WT and $Prkg1^{RQ/+}$ mice ($n = 4$ male mice per genotype), and mean arterial pressure (MAP) averaged separately during rest (7 a.m.–7 p.m.) and activity (7 p.m.–7 a.m.). **e** Ultrasound images of thoracic aortas of 12-month-old WT and $Prkg1^{RQ/+}$ mice. Measurements were performed at end-diastole, indicated by an arrowhead below the ECG (A = aortic root; B = ascending aorta, and C = aortic arch; RPA=right pulmonary artery; LSA=left subclavian artery). **f** Diameter of the aortic root, ascending aorta, and aortic arch in 12-month-old WT and $Prkg1^{RQ/+}$ mice ($n = 6$ M + 7 F mice per genotype). **g–j** Elastin fiber breaks (**g**), SMC density (**h**), SMC apoptosis (**i**), and media collagen content (**j**) of ascending aorta cross-sections from the animals in panel **f** were assessed by Van Gieson's (**g**), hematoxylin/eosin (**h**), TUNEL (**i**), and Masson-Trichrome (**j**) stains (×80 with 25 μm scale bars in **h**, **i** and ×40 with 50 μm bars in **g**, **j**). Arrows in **g** and **i** point to elastin fiber breaks and some TUNEL-positive nuclei, respectively. Collagen content (**j**, blue) of aortic media was quantified using ImagePro (A = adventitia; excluded). In panels **g** and **h** $n = 5$ M + 5 F mice and in panels **i** and **j** $n = 3$ M + 3–4 F per genotype. Graphs show means ± SD; *$p < 0.05$, **$p < 0.01$, ***$p < 0.001$ for the indicated comparisons by two-sided Student's $t$-test (panels **b**, **f**, **h**, **i**, **j**), Mann–Whitney test (**g**), or two-way ANOVA (panels **a**, **d**; #$p < 0.05$ for comparison of basal activity in WT versus RQ/+; +$p < 0.05$ and +++$p < 0.001$ for comparison of the same condition in the absence versus presence of DT2). In panels **g**, **h**, ns = non-signficant. Source data are provided as a Source Data file

**Effects of mutant PKG1$^{RQ}$ in human aortic SMCs.** To study the mechanism whereby the mutant PKG1$^{RQ}$ leads to biochemical changes and changes in gene expression, we expressed the wild type and mutant enzyme in primary human aortic SMCs at levels 2–3-fold above endogenous PKG (Fig. 3a). We found increased basal PKG activity in cells expressing the mutant enzyme—consistent with more of the mutant enzyme present than the endogenous enzyme; basal VASP phosphorylation was increased in PKG1$^{RQ}$-expressing cells, while cGMP-stimulated phosphorylation was similar in cells expressing wild type or mutant enzyme (Fig. 3a, b). Cells expressing PKG1$^{RQ}$ showed increased expression of contractile genes, and TGF-β1 and its target genes, including $Nox4$; expression of these genes was stimulated by TGF-β in both control cells and PKG$^{RQ}$-expressing cells (Fig. 3c, d; Supplementary Fig. 4a). The PKG1$^{RQ}$-expressing cells produced more H$_2$O$_2$ than control cells or cells expressing wild-type PKG1, and shRNA silencing of $Nox4$ reduced H$_2$O$_2$ production and NADPH oxidase activity to control levels; $Nox2$ shRNA had minimal effects (Fig. 3e, f, Supplementary Fig. 4b–d). A NOX1/4 inhibitor (GKT137831)$^{35}$ reduced H$_2$O$_2$ production in the PKG1$^{RQ}$-expressing cells to a level found in control cells (Fig. 3g). These results indicate that NOX4 was the major source of excess H$_2$O$_2$ in cells expressing the mutant kinase. The PKG1$^{RQ}$-expressing cells also showed higher basal and TGF-β-induced JNK activation and DNA and protein oxidation than control cells, recapitulating findings in $Prkg1^{RQ/+}$ aortas (Fig. 3h–j, Supplementary Fig. 4e, f). The PKG1$^{RQ}$-induced increase in DNA oxidation was largely prevented by DT2, indicating a requirement for PKG activity; the pro-oxidant effect of PKG1$^{RQ}$ was mimicked by $Nox4$ over-expression and blocked by GKT137831 (Fig. 3j, k).

PKG1$^{RQ}$ did not appear to regulate the $Nox4$ promoter via TGF-β, because PKG1$^{RQ}$-induced luciferase activity from a $Nox4$ promoter-luciferase reporter was not affected by an inhibitor of TGF-β receptor-1 (ref. $^{36}$), although the drug prevented promoter activation by TGF-β (Supplementary Fig. 4g). However, stimulation of the $Nox4$ promoter required JNK activity and PKG1$^{RQ}$ enhanced the stimulatory effect of c-Jun on the promoter, suggesting that PKG1$^{RQ}$ stimulation of $Nox4$ transcription is mediated by JNK/c-Jun (Supplementary Fig. 4h). Similarly, the oxysterol 7-ketocholesterol increases $Nox4$ transcription in human SMCs via activation of JNK/c-Jun$^{37}$.

Expression of PKG1$^{RQ}$ in the human SMCs inhibited growth factor-induced proliferation and induced apoptosis (Supplementary Fig. 5a–c), consistent with effects of NO/cGMP-induced PKG1 activation in rodent SMCs$^{15,32,38}$. The growth-inhibitory and pro-apoptotic effects of PKG1 were partly prevented by

DT2 and GKT137831, and were mimicked by NOX4 over-expression, suggesting they were in part mediated by NOX4-induced oxidative stress (Fig. 3l–o). Phosphodiesterase-5 inhibitors such as sildenafil increase intracellular cGMP concentrations and activate PKG$^{11}$. In the human aortic SMCs, sildenafil increased VASP phosphorylation and induced JNK activation when combined with low concentrations of an NO donor; the drug combination also inhibited SMC proliferation, thus mimicking some important effects of PKG1$^{RQ}$ (Supplementary Fig. 5d–f).

**Prevention of age-related aortic pathology by cobinamide.** To determine if reducing reactive oxygen species could prevent the pathological changes that occur in the thoracic aortas of $Prkg1^{RQ/+}$ mice, we treated mice with cobinamide (Cbi), a vitamin B$_{12}$ analog that scavenges free radicals and exhibits potent antioxidant properties$^{39}$. Providing Cbi in the drinking water from age 6 to 12 months reduced oxidative stress markers and SMC apoptosis in aortas of $Prkg1^{RQ/+}$ mice to values in WT mice (Fig. 4a–c, Supplementary Fig. 6a). Notably, the drug completely prevented aortic dilation and SMC loss in the mutant mice, and reduced elastin fiber breaks and media collagen content to levels in WT mice (Fig. 4d–f, Supplementary Fig. 6b). In human SMCs, Cbi decreased H$_2$O$_2$-induced JNK activation and protein oxidation, and it reduced protein oxidation in PKG1$^{RQ}$-expressing cells (Supplementary Fig. 6c–e). Cbi had no effect on the activity of purified PKG1$^{RQ}$, nor did Cbi treatment of the mice affect PKG activity in aortic extracts (Supplementary Fig. 7a, b). Thus, excess oxidative stress appears responsible for aortic disease caused by constitutive PKG1 activation.

Cbi had no significant effects in wild-type mice, and no discernable toxicity, as judged by normal weight, clinical appearance, blood counts, and liver and kidney function tests (Supplementary Fig. 7c and Supplementary Tables 1 and 2). Cbi did not affect vitamin B$_{12}$-dependent functions, as indicated by normal serum homocysteine and methylmalonic acid concentrations in Cbi-treated mice (Supplementary Table 1).

Oxidative stress may activate PKG1 in a cGMP-independent fashion, presumably via cysteine 43 oxidation, although the significance of PKG1 redox regulation in vivo is controversial$^{40,41}$. We found a small and similar amount of cysteine 43-oxidized, cross-linked PKG1 dimer in the aortas of wild type and $Prkg1^{RQ/+}$ mice, with no effect of Cbi on dimer amount (Supplementary Fig. 7d). Treating human SMCs with high concentrations of H$_2$O$_2$ (i.e., 100–200 μM) induced cysteine 43 oxidation of PKG1α, but it did not increase VASP

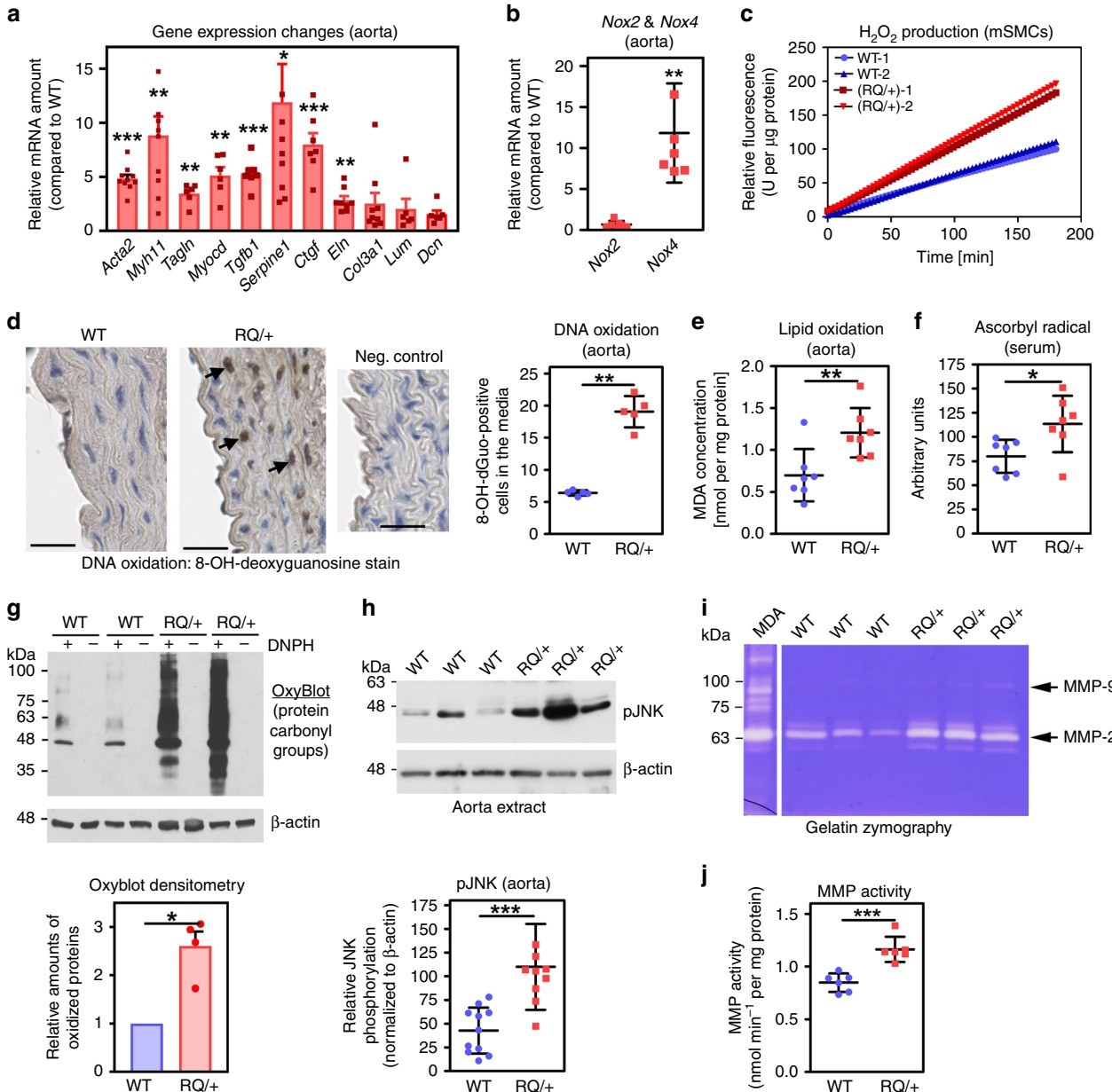

**Fig. 2** Gene expression changes, *Nox4* upregulation, oxidative stress, and increased MMP activity in *Prkg1*[RQ/+] aortas. **a**, **b** Relative mRNA expression in aortas of 4-month-old PKG1[RQ/+] mice compared to wild-type mice: smooth muscle α2-actin (*Acta2*); myosin heavy chain-11 (*Myh11*); transgelin (*Tagln*); myocardin (*Myocd*); TGF-β1 (*Tgfb1*); plasminogen activator inhibitor-1 (*Serpine1*); connective tissue growth factor (*Ctgf*); collagen3-α1 (*Col3a1*); elastin (*Eln*); lumican (*Lum*); decorin (*Dcn*); and NADPH oxidase-2 and -4 (*Nox2*, *Nox4*). Genes of interest were normalized to 18S rRNA and the mean ΔCt of WT mice was assigned a value of one ($n = 3$–4 M + 3–4 F mice per genotype). **c** $H_2O_2$ production of aortic SMCs isolated from WT (in blue) and PKG[RQ/+] (in red) mice measured over time by Amplex Red fluorescence. **d**–**g** Oxidative stress markers measured in aorta and serum of 8- to 12-month-old WT and PKG[RQ/+] mice. DNA oxidation was assessed by immunohistochemical staining of 8-OH-deoxyguanosine (**d**: arrows point to examples of brown nuclei counted as positive; scale bar 50 μm, $n = 2$ M + 3 F mice per genotype). Lipid peroxidation was assessed by measuring aorta malondialdehyde (MDA) content using thiobarbituric acid (**e**, $n = 2$ M + 5 F mice). Ascorbyl radical was measured in serum by electron paramagnetic resonance (**f**, $n = 2$ M + 5 F mice). Protein carbonyl groups were detected by OxyBlot[TM] after derivatization of aortic extracts with 2,4-dinitrophenylhydrazine (DNPH); non-derivatized extracts served as control and reprobing with a β-actin antibody showed equal loading (**g**, ImageJ quantification for $n = 2$ M + 2 F mice per genotype below). **h** JNK activation was assessed on western blots of aortic extracts using a phospho-specific antibody, and was normalized to β-actin (ImageJ quantification for $n = 4$ M + 6 F mice per genotype below). **i**, **j** Metalloproteinase (MMP) activity was assessed in aortic extracts by gelatin zymography (**i**, $n = 1$ M + 2 F mice per genotype) and measured using fluorescently labeled elastin as a substrate (**j**, $n = 2$ M + 4 F mice per genotype; conditioned medium from MDA-MB231 cells served as positive control). Graphs (**a**, **g**) show means ± SEM, other plots means ± SD; *$p < 0.05$, **$p < 0.01$, ***$p < 0.001$ for the indicated comparisons by one-sample *t*-test (panels **a**, **b**, **g**), two-sided *t*-test (**e**, **f**, **h**, **j**), or Mann–Whitney test (**d**). Source data are provided as a Source Data file

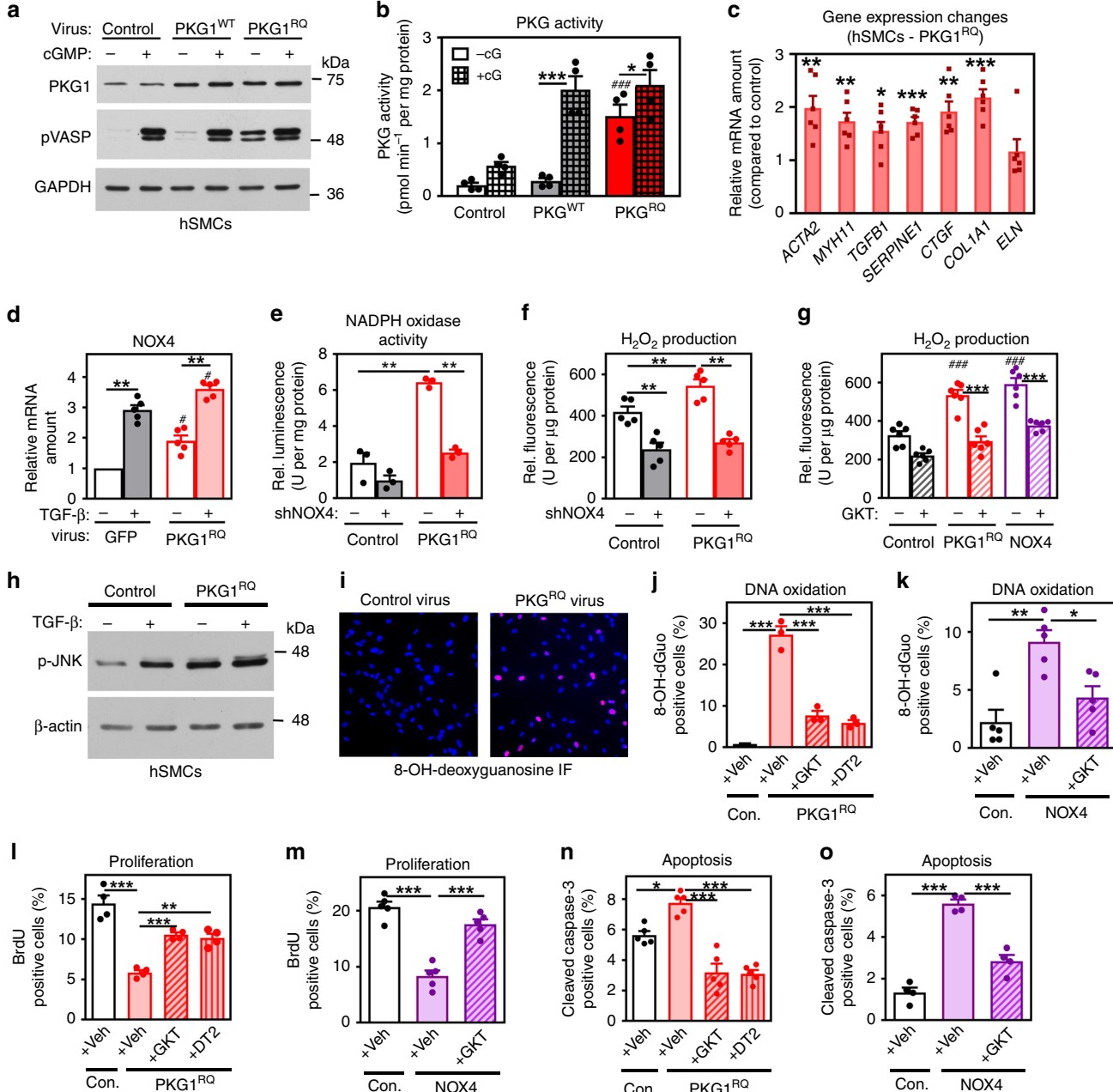

**Fig. 3** PKG$^{RQ}$-induced oxidative stress, apoptosis, and reduced proliferation in human SMCs. Human aortic SMCs were infected with adenovirus encoding green fluorescent protein (control, in black and white), wild-type PKG1 (PKG1$^{WT}$, in **b** gray), PKG1$^{RQ}$ (in red), or NOX4 (in purple), as indicated. (**a, b**) Similar amounts of PKG1$^{WT}$ or PKG1$^{RQ}$ were expressed (**a**). PKG activity was assessed in the absence and presence of cGMP following endogenous VASP phosphorylation in cells (**a**) or using a synthetic peptide (**b**). **c, d** Relative mRNA expression in SMCs expressing PKG1$^{RQ}$ was normalized to phosphoglycerokinase-1 mRNA and compared to cells infected with control virus; some cells (in **d**) were treated with TGF-β for 24 h (gene names as in Fig. 2a, b). **e, f** NADPH oxidase activity and H$_2$O$_2$ production were measured in SMCs infected with control virus or virus encoding shRNA specific for NOX4 (NOX4 mRNA reduction by the shRNA is shown in Supplemental Fig. 4c). **g** H$_2$O$_2$ production was measured in cells infected with virus encoding PKG1$^{RQ}$ or NOX4, and some cells were treated with the NOX1/4 inhibitor GKT137831 (GKT). **h** JNK activation was assessed in SMCs treated with vehicle or TGF-β. **i–k** DNA oxidation was assessed by immunofluorescence staining for 8-OH-deoxyguanosine (**i**: pink nuclei; DNA was counterstained with Hoechst 33342). Some cells were treated with GKT137831 or with the PKG inhibitor DT2. **l, m** SMC proliferation was assessed by Br-deoxyuridine (BrdU) uptake into S-phase nuclei, with some cells treated with GKT137831 or DT2. **n, o** Apoptosis was assessed by immunofluorescence staining for cleaved caspase-3 of cells cultured in 0.5% FBS. Graphs show means ± SEM of three (**e, j**), four (**b, l, o**), five (**d, f, k, m, n**), or six (**c, g**) independent experiments. *$p < 0.05$, **$p < 0.01$, ***$p < 0.001$ for the indicated comparisons; #$p < 0.05$, ##$p < 0.01$, ###$p < 0.001$ for the comparison between PKG1$^{RQ}$- or NOX4-expressing cells versus control cells under the same condition (panel **c** by one-sample $t$-test, panels **j–o** by one-way ANOVA, and panels **a** and **d–g**, by two-way ANOVA). Source data are provided as a Source Data file

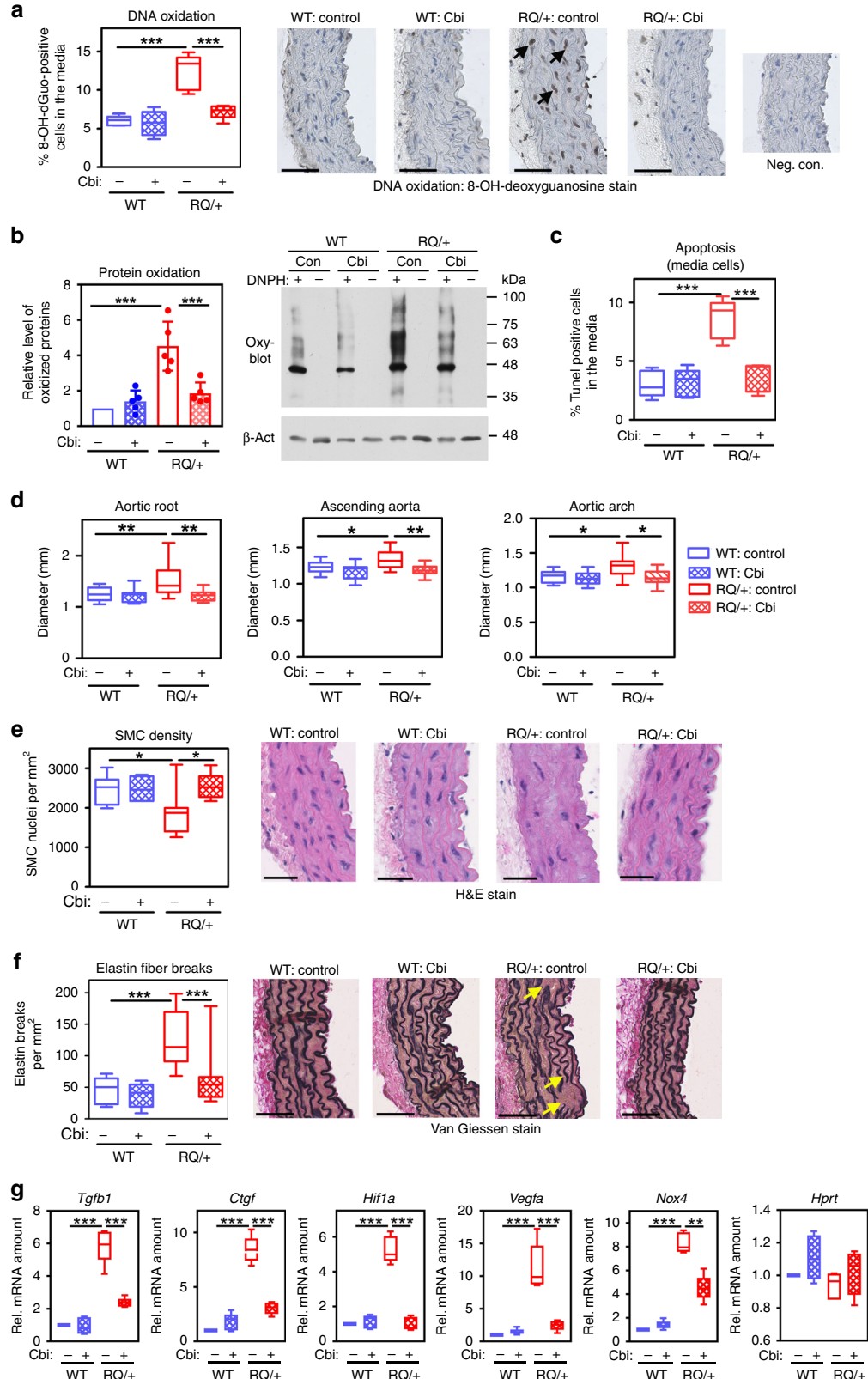

phosphorylation, confirming our previous observation that cysteine 43 oxidation does not activate the kinase (Supplementary Fig. 7e)[41].

Gene expression profiling in *Nox4*-transgenic and -knockout mice previously identified *Tgfb1* and hypoxia-inducible factor-1α (*Hif1a*) as potential downstream targets of *Nox4*[42–44]. We found increased expression of *Nox4*, *Tgfb1* and its target *Ctgf*, as well as *Hif1a* and its target vascular endothelial growth factor-A (*Vegfa*) in the aortas of 12-month-old *Prkg1*$^{RQ/+}$ mice compared to wild-type littermates (Fig. 4g). Cbi treatment of the mutant mice reduced all five transcripts, with *Hif1a* and *Vegfa* mRNAs reaching levels similar to those observed in wild-type mice

**Fig. 4** Reduction of oxidative stress and prevention of age-dependent aortic pathology in cobinamide-treated PKG$^{RQ/+}$ mice. Starting at 6 months of age, wild type (WT, in blue, $n = 5$ M + 4 F mice) and *Prkg1*$^{RQ/+}$ mice (RQ/+, in red, $n = 5$ M + 7 F mice) received 1 mM histidyl-cobinamide (Cbi, cross-hatched bars) in the drinking water for 6 months; controls were left untreated (open bars, WT $n = 7$ M + 7 F, and RQ/+$n = 9$ M + 11 F mice). All mice were analyzed by ultrasound (**d**); at least five mice per group (both genders) were analyzed in the other panels. **a** DNA oxidation assessed by immunohistochemical staining with an antibody for 8-OH-deoxyguanosine; brown nuclei in the medial layer were counted (arrows show examples of cells counted as positive). **b** Protein carbonyl groups detected by OxyBlot™ in aortic extracts as described in Fig. 2g and quantified by densitometric scanning. **c** SMC apoptosis assessed by TUNEL staining; only TUNEL-positive nuclei in the media were counted. **d** Diameter of the aortic root, ascending aorta, and aortic arch were measured by ultrasound at the age of 12 months. **e** SMC density measured by counting SMC nuclei on hematoxylin/eosin-stained sections. **f** Elastin fiber breaks (arrows show examples) quantified on Van Giessen-stained cross-sections of the ascending aorta. Scale bars are 25 μm. **g** mRNA expression in aortas of 12-month-old mice treated for 1 month with 1 mM Cbi in the drinking water (or left untreated, five female mice per group): TGF-β1 (*Tgfb1*); connective tissue growth factor (*Ctgf*); hypoxia-inducible factor-α (*Hif1a*); vascular endothelial growth factor-A (*Vegfa*); NADPH oxidase-4 (*Nox4*), and hypoxanthine phosphoribosyltransferase (*Hprt*). Genes were normalized to 18S rRNA and the mean ΔCt of WT mice was assigned a value of one. Boxplots show median and interquartile range with whiskers indicating the total range, and the graph in **b** shows means ± SD (*$p < 0.05$, **$p < 0.01$, and ***$p < 0.001$ for the indicated comparisons by two-way ANOVA). Source data are provided as a Source Data file

(Fig. 4g). Cbi suppression of *Nox4* mRNA in the mutant mice may be explained by redox regulation of the *Nox4* promoter[45]. Since dysregulation of the *Hif-1α/Vegfa* and *Tgfb1/Ctgf* pathways has been linked to aortic aneurysm progression[9,28,46,47], these *Nox4* target genes likely contribute to aortic pathology in PKG1$^{RQ/+}$ mice.

**Transverse aortic constriction-induced aortic rupture in young *Prkg1*$^{RQ/+}$ mice.** To determine if increased wall stress affected the aortas of young *Prkg1*$^{RQ/+}$ mice, we subjected 4-month-old mice to transverse aortic constriction (TAC)[48], which increased blood pressure in the ascending aorta in WT and *Prkg1*$^{RQ/+}$ similarly (Fig. 5a, Supplementary Fig. 8a, b). Within 2 weeks of TAC, 36% of *Prkg1*$^{RQ/+}$ mice succumbed to aortic rupture—with hemothorax found on necropsies—while none of the WT mice died (Fig. 5b, Supplementary Fig. 8c). Surviving mice of both genotypes showed similar pre-stenotic aortic dilation (Fig. 5c, Supplementary Fig. 8d). TAC induced severe aortic wall pathology in young *Prkg1*$^{RQ/+}$ mice, with thinning of the media, elastin fiber breaks, loss of SMCs, increased SMC apoptosis, DNA oxidation, and media collagen content; none of these changes occurred in aortas from WT mice (Fig. 5d–i, Supplementary Fig. 8e–f). To assess if oxidative stress contributed to the TAC-induced changes in the *Prkg1*$^{RQ/+}$ mutants, we treated mice with the antioxidant *N*-acetylcysteine (NAC) for 4 weeks prior to and 2 weeks after TAC. We purposefully chose an antioxidant that was structurally and mechanistically unrelated to Cbi. NAC treatment largely prevented TAC-induced aortic pathological changes, including media thinning, and reduced oxidative stress in *Prkg1*$^{RQ/+}$ mice, with no obvious effect in wild type mice (Fig. 5d–i, Supplementary Fig. 8e, f). The survival of NAC-treated mutant mice subjected to TAC did not differ from that of untreated mice, and NAC did not affect pre-stenotic aortic dilation or aortic pressure gradients after TAC surgery.

## Discussion

We identified several mechanisms whereby constitutive PKG1 activation induces thoracic aortic pathology (Fig. 5j): (i) JNK is activated, up-regulating NOX4, and leading to increased oxidative stress, which further increases JNK activation; (ii) oxidative stress and JNK activation initiate SMC apoptosis and MMP-2 activation, causing elastin fiber degradation; (iii) impaired SMC proliferation and increased SMC apoptosis lead to SMC depletion; and (iv) increased TGF-β expression and target gene upregulation may contribute to aortic pathology, at least in late stages of aneurysm progression[9,28,49]. NOX4 upregulation leads to increased *Hif-1α* and *Vegfa* expression, and dysregulated HIF-1α /VEGF signaling has been observed in TAAD and linked to aortic aneurysm progression[43,44,46,47,50,51].

In contrast to decreased SMC contractile proteins typically found in familial TAAD[2,5,6,52], PKG1$^{RQ}$-expressing SMCs showed an increase in contractile markers, consistent with the identified role of PKG1 in SMC phenotypic modulation[12–15]. Our data suggest that PKG1$^{RQ}$-expressing SMCs are locked in a low proliferative, contractile state, which interferes with vascular repair and aortic wall homeostasis.

Increased reactive oxygen species and JNK activation leading to MMP upregulation have been implicated in several types of TAAD and abdominal aortic aneurysms[7,8,31,52–54]. However, anti-oxidants and MMP inhibitors have had disappointing results in treatment of aortic aneurysms, which may be because of low systemic drug concentrations and/or low target affinities[1,55,56]. Cbi was highly effective in preventing age-related aortic pathology, including aortic dilation in the *Prkg1*$^{RQ/+}$ mice, and it reduced oxidative stress markers in the aorta. Cbi reacts very quickly with $O_2^-$ and has a favorable toxicity profile[39], suggesting it could retard onset of other forms of TAAD associated with oxidative stress[52,53,57]. NAC was effective in ameliorating aortic wall injury after TAC, but its effect on mortality was limited, perhaps due to mechanical factors contributing to aortic rupture. Studies to confirm a role of NOX4 in aortic disease could be pursued by crossing *Prkg1*$^{RQ/+}$ mice with *Nox4* knockout mice; however, *Nox4*-deficient mice exhibit vascular abnormalities— including endothelial dysfunction and apoptosis—which would confound interpretation of results[58,59]. Basal NOX4-derived $H_2O_2$ appears to have some vascular-protective effects, while increased NOX4 expression can exert both adverse and protective actions in cardiovascular disease models[43,60–62].

Our data identify constitutive PKG1 activation as a cause of increased oxidative stress and vascular damage. The clinical significance of this finding reaches beyond the small number of people carrying the activating *PRKG1*,p.Arg177Gln mutation, because widespread use of PKG1-activating agents for a variety of disorders[10] may have unexpected long-term consequences on aortic wall homeostasis.

## Methods

**Generation of PKG1$^{RQ}$ knock-in mice.** All animal experiments complied with ethical guidelines for the use of animals in research according to policies of the University of California, and were approved by the Institutional Care and Use Committee of the University of California, San Diego. Mice were housed in groups of 2–4 in a temperature-controlled environment with 12/12 h light/dark cycle. To generate PKG1$^{RQ}$ knock-in mice, we used KOD Xtreme Hotstart polymerase (EMD Millipore Corporation, Billerica, MA, USA) with 129S1 embryonic stem cell genomic DNA as a template to amplify *prkg1* exon III, a 3.4-kb fragment of 5′-flanking sequence, and a 3.1-kb 3′-flanking sequence. The arginine to glutamine mutation was generated in exon 3 by overlapping extension PCR using the primers originally used to amplify the 3′ fragment, and the following mutagenesis primers: 5′-CTGTACCCAGACAGCGACCGTCAAGAG-3′ and 5′-GGTCGCTCTCTGG GTACAGTTGTAAAG-3′. A targeting construct was assembled with the fragments

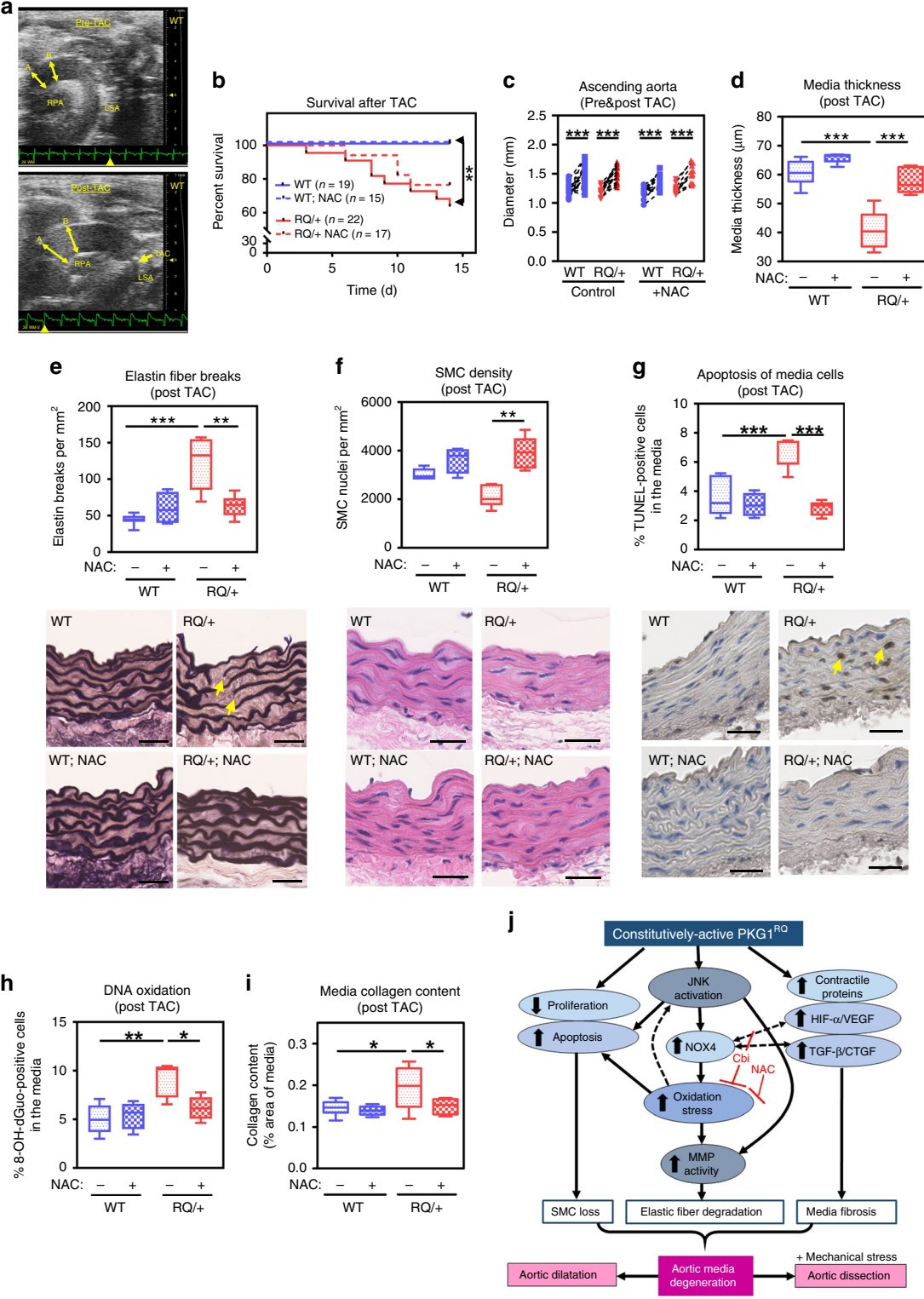

on either side of a Frt-flanked neomycin cassette (Supplementary Fig. 1b), and all PCR products were sequenced. The linearized construct was electroporated into 129S1 embryonic stem cells, and G418-resistant clones were screened by PCR using the primers 5′-ATCCTCATCCCCACAACAG-3′ and 5′-TCTTTTGTGACTTCA ACCTTCC-3′. Homologous recombination was confirmed by Southern blot analysis of positive clones, using probes outside of the targeting construct (Supplementary Fig. 1b, c). A positive clone with normal chromosome analysis in 20 metaphase spreads was injected into C57Bl/6 blastocysts to establish chimeric mice. Male chimeric mice were mated to C57BL/6J females and agouti pups were screened for the presence of the mutated allele. To remove the neomycin cassette, heterozygous mice were mated with flip recombinase-expressing mice in a 129S4

background (JAX mouse stock #003946). The removal of the neomycin cassette was confirmed, and subsequent genotyping was performed by PCR using tail DNA template with the following primers: 5′-ATAATGCCCAAGGGTTGGAG-3′ and 5′-AAAGGCTGGGGAAAGGTAGG-3′ (primers F1/R1 in Supplementary Fig. 1d). Additional screening was performed by restriction fragment length polymorphism; genomic DNA was PCR amplified with the primers 5′-AAGCTCCACGTTTTC AGC-3′ and 5′-GATATCATCCCCAGGCTTTTG-3′, and the 531 bp PCR product was digested with NciI. The wild-type allele is cut into 188 and 343 bp fragments and the mutant allele remains uncut (Supplementary Fig. 1e). To produce mice in an inbred 129S background, offsprings from matings with 129S4 flip recombinase-expressing females were backcrossed for at least three generations with

**Fig. 5** TAC-induced severe aortic pathology and death from aortic rupture in young $Prkg1^{RQ/+}$ mice. **a** Aortic arch by ultrasound imaging before and 2 weeks after TAC in a wild-type mouse (A = aortic root; B = ascending aorta; RPA=right pulmonary artery; LSA=left subclavian artery). **b** Survival of male wild type (WT, in blue) and PKG1$^{RQ/+}$ (RQ/+, in red) mice after TAC ($n = 19$ WT and $n = 22$ RQ/+). Some animals received $N$-acetylcysteine (NAC) for 4 weeks before and 2 weeks after TAC ($n = 15$ WT and $n = 17$ RQ/+). Only mice surviving 24 h after surgery were included, euthanized 14 days after surgery. **$p < 0.01$ by log-rank (Mantel–Cox) test for the indicated comparison. **c** Diameter of ascending aortas before and 14 days after TAC, measured in surviving WT and $Prkg1^{RQ/+}$ mice ($n$ as in panel **b**, ***$p < 0.001$ by two-way ANOVA). **d** Media thickness measured on cross-sections of ascending aortas in surviving mice. **e–g** Elastin fiber breaks, SMC density, and apoptosis of media cells assessed on ascending aorta cross-sections of mice euthanized 14 days after TAC surgery (scale bars 25 μm; arrows show examples of elastin fiber breaks and TUNEL-positive nuclei). **h** DNA oxidation assessed by staining for 8-OH-deoxyguanosine in nuclei of media cells. **i** Media collagen content assessed on Masson–Trichrome stains. Boxplots in **d–i** show median and interquartile range with whiskers indicating the total range, for $n = 6$ male mice per group; *$p < 0.05$, **$p < 0.01$, ***$p < 0.001$ for the indicated comparisons by two-way ANOVA. Source data are provided as a Source Data file. **j** Schema summarizing consequences of mutant PKG1$^{RQ}$ expression in the aorta. PKG1 activates JNK causing increased H$_2$O$_2$ production from NOX4 upregulation; oxidative stress further enhances JNK activation, activates and upregulates MMPs via JNK, and MMPs degrade elastin fibers. PKG1 activation causes SMC loss, in part via JNK activation and oxidative stress. PKG1 increases expression of contractile proteins, HIF-1α/VEGF, and TGF-β/TGF-β target genes; the latter promote aortic media fibrosis, which may be compensatory. Aortic media degeneration leads to age-dependent aortic dilation, and aortic dissections under hypertensive stress. Cobinamide (Cbi) or $N$-acetylcysteine (NAC) treatment reduce oxidative stress and JNK activation, preserving aortic media integrity

129S1/SvImJ mice (JAX mouse stock #002448). These mice were used for all experiments, except in the TAC experiments shown in Fig. 5 and Supplementary Fig. 8. Mice used for TAC experiments were backcrossed for at least eight generations into a C57BL/6NHsd background (Envigo catalog #044); this was done because cardiac outcomes after TAC have been well characterized in this background, and in male mice there is no evidence of heart failure 14 days after TAC[48]. Except for TAC, all experiments were performed with both male and female littermates from multiple litters, and results are shown for mixed gender, unless stated otherwise.

**Randomization and drug treatment of mice**. At the time of weaning, male and female littermates were separated and randomly assigned to new cages. At the age of 6 months (aging studies) or 10–14 weeks (TAC studies), matched cages containing mice born within a 3-week period were randomly assigned to treatment versus vehicle groups. For aging studies, some mice received 1 mM trihistidyl-cobinamine (Cbi) in the drinking water, starting from the age of 6 months until 12 months. For gene expression studies, mice were treated with 1 mM Cbi for one month only, from 11 to 12 months of age. For TAC studies, some mice received 1 mg/mL NAC in the drinking water starting 4 weeks before until 2 weeks after the surgery. The drugs were dissolved freshly each week and filter-sterilized; fluid intake was monitored weekly. The Cbi dose was calculated to be 50 mg/kg/day, and the NAC dose was 500 mg/kg/day. Aquohydroxyl–cobinamide was synthesized from cobalamine by removal of the 5,6-dimethyl-benzimidazol ring; trihistidyl-cobinamide was obtained by adding three molar equivalents of L-histidine to the aquohydroxyl-cobinamide[63]. Purity of the drug was >95% by high-pressure liquid chromatography.

**Telemetry blood pressure recordings**. Blood pressures were measured on conscious 4-month-old mice using a telemetry blood pressure recording system (Data Sciences International, Ponemah; PA-C10 pressure transmitter). Mice were anesthetized with isoflurane (1.5%), and a 20–25 mm midline incision was made vertically at the ventral throat area. The PAC-10 transmitter catheter was introduced into the left carotid artery and the tip was advanced into the thoracic aorta. Data were recorded for 3 days, after a 2-week post-operative recovery period.

**Transthoracic echocardiography**. Prior to echocardiography, a depilatory cream was applied to the anterior chest wall to remove hair. Mice were anesthetized with 5% isoflurane for 15 s and then maintained at 0.5% throughout the echocardiography examination. Small needle electrodes for simultaneous electrocardiogram were inserted into one upper and one lower limb. Transthoracic echocardiography was performed using the FUJIFILM VisualSonics Inc., Vevo 2100 high-resolution ultrasound system with a linear transducer of 32-55 MHz. Measurements of aortic diameters were recorded during end-diastole at the aortic root, ascending aorta, and aortic arch; they were performed by a single, highly experienced operator, who was blinded to genotype and treatment of the mice. Bland Altman analyses were performed to determine inter- and intra-observer variability of echo measurements (Supplementary Fig. 9). Measuring the diameter of the ascending aorta, the average of the differences between two observers was −0.061 mm (with 95% limits of agreement between −0.18 and 0.06 mm), and the average of differences between two observations by the same observer were 0.024 mm (with 95% limits of agreement between −0.12 and 0.17 mm).

**Arterial ring isometric tension measurements**. Isometric tension in aortas was measured as previously described[64], with minor modifications. Thoracic aortas were dissected in Krebs–Henseleit solution, cleaned of any adherent connective tissue, and cut into 1.5–2 mm segments. Rings with intact endothelium were mounted in a wire myograph over 40 μm wires, set at a resting tension of 0.5 g, and allowed to equilibrate at 37 °C for 45 min with intermittent washes every 15 min. After equilibration, aortas were treated with 40 mM K$^+$-solution. For vascular relaxation by acetylcholine (Sigma) or 8-(4-chlorophenylthio)cGMP (8-CPT-cGMP; BioLog), each aortic ring was pre-contracted to generate ~0.15 g contraction by treatment with prostaglandin –F2α (PGF$_{2α}$ 2–8 μM; Thermo-Fisher). The degree of relaxation was calculated as a percent of PGF$_{2α}$-induced contraction. Reagent sources and catalog numbers are summarized in Supplementary Table 3.

**PKG1 purification and activity assays**. Wild type and mutant PKG1$^{RQ}$ were expressed with N-terminal FLAG epitope tags in transiently transfected 293T cells (ATCC, CRL3216), purified with anti-FLAG beads and eluted with FLAG peptide, and kinase activity assays were performed[41]. Kinase activity in SMC and aortic extracts was measured using an optimized peptide (TQAKRKKSLAMA, 30 μM) in the presence of 1.25 μM protein kinase inhibitor peptide (PKI) to inhibit cAMP-dependent protein kinase[65]. Assays were performed with [γ-$^{32}$PO$_4$]ATP (1.3 Ci/mmol; 50 μM) in the presence and absence of 3 μM cGMP, and were linear with time and protein concentration. In some assays, the PKG-specific inhibitory peptide DT2 was included at 10 μM.

**Western blotting**. Following anesthesia with ketamine (100 mg/kg) and xylazine (10 mg/kg) by intraperitoneal injection, mice were exsanguinated and perfused with ice-cold phosphate-buffered saline (PBS) containing protease inhibitor cocktail (Calbiochem #539131). Aortas were excised, cleaned from surrounding tissue, and snap-frozen in liquid nitrogen within 15 min of euthanasia. Frozen aortas were pulverized and sonicated ($3 \times 10$ s) in ice-cold RIPA buffer (50 mM NaCl, 50 mM Tris-HCl, pH 8, 1% NP-40, 0.1% SDS, 0.5% sodium deoxycholate) supplemented with 1 mM NaF, 1 mM β-glycerolphosphate, and protease inhibitor cocktail. Western blots were developed with primary antibodies specific for PKG1 (1:1000; Cell Signaling Technology; 3248, see Supplementary Table 4), VASP-phospho-Ser$^{259}$ (1:1000; Cell Signaling Technology; 3114), GAPDH (1:1000; Cell Signaling Technology; 2118), β-actin (1:5000; Santa Cruz Biotechnology; 47778), and JNK-phospho-Thr$^{183}$/Tyr$^{185}$ (1:1000; Cell Signaling Technology; 9251). Blots were developed and visualized with enhanced chemiluminescence using LI-COR Odyssey (with ImageStudio, V5) or with film in the linear range of exposure (with densitometry scanning using ImageJ (V1.51). Protein carbonyl groups (aldehydes and ketones) were detected using the OxyBlot$^{TM}$ Protein Oxidation Detection Kit (Millipore EMD; S7150). Unprocessed scans of blots are provided in the Source Data file.

**Histomorphometric studies**. Following anesthesia with i.p. ketamine (100 mg/kg) and xylazine (10 mg/kg), mice were perfusion-fixed with 10 mL of 4% paraformaldehyde for 3 min under physiological pressure. Thoracic aortas were excised and further fixed overnight. Paraffin-embedded ascending aortas were cross-sectioned into 5-μm-thick sections, which were stained with haematoxylin and eosin (to count SMC nuclei), Van Gieson elastin stain (to detect elastin fiber breaks), and Masson's Trichrome stain (to quantify collagen), following standard protocols. Slides were scanned with a Hamamatsu NanoZoomer 2.0 HT System and analyzed using Digital Pathology NDP.view2 software. The images were captured at ×10, ×40, and ×80 magnification for histomorphometric analyses[66]. The thickness of the aortic media was measured using a straight-line tool at four different points (at 0°, 90°, 180°, 270°) on two different sections, and the mean was reported. SMC nuclei and elastin fiber breaks were counted manually at ×40 magnification, on five non-overlapping areas of aortic media measuring 0.025 μm$^2$ each; means were calculated and expressed per mm$^2$. Collagen content of the media area was measured at ×10 magnification on Masson's Trichrome-stained cross-sections using ImagePro Premier software (V 9.0, Media Cybernetics).

Histomorphometric measurements were confirmed by an investigator who was blinded to the genotype and treatment group of the mice.

**Immunohistochemical staining of aortic sections.** Terminal deoxynucleotidyl transferase-mediated dUTP-biotin nick end-labeling (TUNEL) of deparaffinized aortic sections was performed using the Apoptag Peroxidase In-situ Apoptosis Detection Kit (Millipore EMD; S7100) according to the manufacturer's instructions. For 8-OH-dGuo staining of deparaffinized sections, antigen retrieval was done by placing slides into boiling 10 mM sodium citrate buffer, pH 6, and letting them cool for 30 min. The sections were treated with RNase A (100 μg/mL) in 10 mM Tris-HCl, pH 7.5, 1 mM EDTA, 0.4 M NaCl at 37 °C for 1 h, rinsed, treated with proteinase K (10 μg/mL) in phosphate-buffered saline (PBS) at room temperature for 7 min, and permeabilized with 0.2% Triton X-100 for 15 min. Samples were blocked in 2% BSA/PBS, and incubated in anti-8-OH-dGuo (1:50 dilution in 1% BSA/PBS), followed by a goat anti-mouse horseradish peroxidase-coupled secondary antibody (1:100; Jackson Immunoresearch). Slides were developed using 3,3-diaminobenzidine substrate (Vector Laboratories) and counterstained with methyl green.

**Dihydroethidium staining of aortas.** Ascending aortas were dissected, and immediately immersed in 10 μM dihydroethidium (Life Technologies; C10422) solution in PBS and incubated for 30 min at 37 °C. The tissue was rinsed ×4 in PBS, placed in Tissue-Tek OCT compound (Miles Laboratories), and snap-frozen in liquid nitrogen. Cryosections were fixed in 4% paraformaldehyde for 15 min at room temperature, washed in PBS, and counterstained in Hoechst 33342; sections were imaged within 4 h by fluorescence microscopy.

**Quantitative RT-PCR.** Aortas were dissected and snap-frozen as described above; they were pulverized and immersed in Trizol (Molecular Res.Center, TR118). Total RNA was isolated, reverse-transcribed using iScript cDNA synthesis kit (Bio-Rad), and PCR was performed using a MX3005P real-time PCR detection system with Brilliant II SYBR Green Mix (Agilent Technologies) as described[67]. Primer sequences are in Supplementary Tables 5 and 6. All primers were intron-spanning (except for 18S rRNA), and were tested with serial cDNA dilutions. Relative changes in mRNA expression were analyzed using the comparative $2^{-\Delta\Delta Ct}$ method, with 18S rRNA and phosphoglycerate kinase-1 serving as internal controls[68].

**SMC culture and adenovirus infection.** Murine aortic SMCs were isolated from the aortas of 8-12 week-old wild type and PKG1$^{RQ/+}$ mice and were cultured in Smooth Muscle Basal Medium (SmBM) containing 20% FBS, pyruvate, HEPES, L-glutamine, penicillin/streptomycin, and growth factors (SmBM Bullet kit; CC-4149 from Lonza), as previously described[69]. Two independent cell isolates from wild type and PKG1$^{RQ}$ mice were used at passage one. Human primary aortic SMCs were purchased from Lifeline Cell Technology (Frederick, MD, FC-0015) and cultured in VascuLife Basal Medium supplemented with recombinant epidermal growth factor, insulin, and fibroblast growth factor-b, ascorbic acid, L-glutamine, 20% FBS, genta-mycin, and amphotericin (Vasculife SMC LifeFactors kit; LS-1040 from Lifeline Technologies). Human SMCs were used at passages 3–6. Adenovirus was generated using the ViraPower Adenoviral Expression System (Thermo Fisher)[67]. Human SMCs were infected with virus expressing wild type or mutant human PKG1α (containing the R177Q mutation); the MOI was 3–10 and titrated to produce wild type and mutant kinase at 2–3-fold the level of endogenous PKG1 at 72 h post infection. For Nox4 over-expression, human SMCs were infected with virus expressing murine Nox4 to increase H$_2$O$_2$ production to a level comparable to that induced by PKG1$^{RQ}$. For knock-down of NOX2 and NOX4, SMCs were infected with adenovirus expressing shNOX2 or shNOX4 oligomers (downstream of the U6 promoter) at an MOI of 100; H$_2$O$_2$ production and NADPH oxidase activity assays were performed 48 h later (as described below).

**Nox4 promoter activity assay.** A 1707bp fragment of the Nox4 promoter region 5′ to the translational start site was cloned from mouse genomic DNA using KOD Hot Start DNA Polymerase (Millipore) and the following primers: 5′-GCTCTC GAGCTTTGCAGGCTCAGGCTC-3′ and 5′-GGTCCATGGCGCCGGCGCGGG GAGTGCT-3′. The PCR product was inserted into pGL3 (Promega) using XhoI and NcoI sites, and sequenced. C3H/10T1/2 cells (ATCC, CCL226) were transfected with lipofecamine 2000, and luciferase activity was measured using a single vial luminometer[13].

**SMC immunofluorescence staining.** SMCs were cultured on glass coverslips and fixed in 4% paraformaldehyde. For 8-hydroxydeoxyguanosine (8-OH-dGuo) staining, cells were treated with RNase A and proteinase K as described above for tissue sections. Cells were permeabilized with 0.2% Triton X-100 in PBS and blocked with 2% BSA in PBS (8 min and 1 h, respectively, at room temperature). Cells were incubated with the anti-8-OH-dGuo antibody (1:100; Abcam, ab26842) in 1% BSA in PBS overnight at 4 °C. Apoptotic cells were stained with anti-cleaved caspase-3 antibody (1:100; Cell Signaling Technology #9664) in 1% BSA in PBS overnight at 4 °C. To label cells in the S phase, cells were incubated with 200 μM 8-

Br-deoxyuridine (BrdU; Sigma) for 48 h, fixed in paraformaldehyde, and per-meabilized with 0.2% Triton X-100 as above. After rinsing with PBS, cells were incubated with DNAse I (Sigma) for 30 min at 37 °C and blocked with 2% BSA in PBS for 1 h, prior to incubation with anti-BrdU antibody (1:200, Sigma) for 1 h at room temperature. After washing in PBS, cells were incubated with a Texas Red-conjugated secondary antibody (1:100; Jackson Immunoresearch), and nuclei were counterstained with Hoechst 33342 (Thermo Fischer Scientific). Images were analyzed with a Keyence BZ-X700 fluorescence microscope.

**Amplex Red and NADPH oxidase assays.** Primary murine SMCs isolated from wild type or PKG1$^{RQ/+}$ aortas, and human SMCs infected with control, wild type, or mutant PKG1$^{RQ}$ virus, were seeded at $4 \times 10^4$ cells/well in 96-well plates, and H$_2$O$_2$ production was measured using an Amplex Red/Hydrogen Peroxide/Per-oxidase Assay Kit (Thermo Fischer Scientific). Cells were incubated 16 h later in 0.1 mL of reaction mixture containing 50 μM Amplex Red and 0.1 units/mL horseradish peroxidase. Fluorescence was measured every min in a BioTek Synergy 2 plate reader, using 540 nm excitation and 590 nm emission wavelengths.

To measure NADPH oxidase activity by lucigenin-enhanced chemiluminescence, SMCs were homogenized in PBS with 1 mM EDTA and protease inhibitor cocktail (Calbiochem #539131), and supernatants were obtained after centrifugation at 750g for 5 min. The lucigenin assay was performed in 50 mM phosphate buffer, pH 7.0, 1 mM EGTA, 150 mM sucrose, with 5 μM lucigenin, and was started by the addition of 100 μM NADPH[52]. Photon emission was measured every second for 5 min in a luminometer, and activity (after subtraction of a buffer blank) was expressed as relative luminescence units per mg protein.

The following NOX2 and NOX4 shRNAs were expressed from adenoviral vectors: shNox2 (sense) 5′-GATCGAG-TGGTGTGTGAATGCCAGATTCA AGAGATCTGGCATTCACACACCACTCTTTTTTG-3′; shNox2 (antisense) 5′-AATTCAAAAAAGAGTGGTGTGTGAATGCCAGATCTCTTGAATCTGG CATTCACACACCACTC-3′. shNox4 (sense) 5′-GATCGGAACAAGTGCAATT TCTAAGTTCAAGAGACTTAGAAATTGCACTTGTTCCTTTTTTG-3′; shNox4 (antisense) 5′- AATTCAAAAAAGGAACAAGTGCAATTTCTAAGTCTCTTGA ACTTAGAAATTGCACTTGTTCC -3′. Amplex Red and NADPH oxidase activity assays were performed 48 h after infection of human SMCs.

**Thiobarbituric acid-reactive substances assay.** Frozen aortas were pulverized and lysed in 0.15 mL ice-cold RIPA buffer and sonicated three times for 10 s. To 0.1 mL of the tissue lysate, 0.2 mL 10% trichloroacetic acid and 0.3 mL 0.67% thiobarbituric acid were added, and samples were boiled for 45 min. The thio-barbituric acid adducts were extracted in butanol and measured using 515 and 553 nm as excitation and emission wavelengths, respectively. Malondialdehyde stan-dards were from Cayman Chemical.

**Ascobyl radical measurement by electron paramagnetic resonance.** Murine blood samples were obtained by cardiac puncture at the time of euthanasia, and serum samples were kept frozen at −80 °C for up to 4 weeks prior to analysis. EPR spectra were recorded at room temperature using a MiniScope MS400 spectrometer (Magnatech). EPR conditions were: modulation amplitude, 0.2 mT; sweep time, 20 s; sweep rate, s$^{-1}$. Peak amplitude was measured in arbitrary units.

**Matrix metalloproteinase activity assays.** Frozen aortas were pulverized and lysed in 10 mM sodium cacodylate, 150 mM NaCl, 10 mM CaCl$_2$, 1 mM ZnCl$_2$, 1% Triton X-100, 0.1% SDS, 0.5% sodium deoxycholate, 0.02% sodium azide, and 2% DMSO. For zymography, 6 μg of extract protein were subjected to SDS-polyacrylamide gel electrophoresis under non-reducing conditions in gels con-taining 1% gelatin[16]. Gels were washed three times in 2.5% Triton X-100 for 30 min at room temperature, and incubated in 50 mM Tris-HCl, pH 7.5, 10 mM CaCl$_2$, 200 mM NaCl, and 1 μM ZnCl$_2$ for 18–36 h at 37 °C. Gels were stained with Coomassie Blue and destained until clear bands showed zones of gelatinolytic activity. Culture supernatant from MDA-MB231 breast cancer cells served as a positive control. Total MMP activity in extracts of aortic arches was also measured with a fluorescently labeled elastin peptide, using the Fluorimetric Sensolyte 520 Generic MMP assay kit (Anaspec; AS-71158) according to the manufacturer's instructions, in the linear range of the assay. Results were confirmed using a MMP Activity Assay Kit from Abcam (ab112146) based on a FRET peptide as MMP activity indicator.

**Transverse aortic constriction.** The TAC procedure was performed on 14-20-week-old wild type and PKG1$^{RQ/+}$ mice by a single, highly experienced operator, who was blinded to genotype and treatment of the mice[48]. Mice were anesthetized with ketamine (50 mg/kg) and xylazine (5 mg/kg) by intraperitoneal injection and then received isoflurane (0.75–1.5%) for complete induction of anesthesia. Mice were ventilated with a pressure ventilator. The chest cavity was entered in the second intercostal space at the left upper sternal border and the transverse aorta was isolated between the carotid arteries. Aortic constriction was performed by tying a 7-0 silk suture ligature against a 27–27.5-gauge needle (according to the body weight), and the needle was promptly removed to yield a constriction of about 0.4 mm in diameter. Following the constriction procedure, the chest was

closed with 6-0 silk sutures. Buprenorphine (0.1 mg/kg, 100 μl/mouse) was given 15–30 min prior to anticipated recovery, and every 12 h for 3 days post-operatively. Mice were euthanized 14 days after the procedure. Mice dying within 24 h after TAC surgery were excluded from analyses: 6/40 wild type and 5/44 heterozygous mice died for a total peri-operative mortality of 13%.

**Blood pressure gradient measurements post TAC**. To evaluate the stress level generated by TAC, the pressure gradient between the two carotid arteries was measured at the end of study (d14). Anesthesia was induced, and mice were ventilated as described for TAC. Both carotid arteries were exposed and cannulated with stretched PE 50 catheters connected to fluid-filled transducers. Both carotid artery pressures were simultaneously recorded and analyzed in LabChart (ADInstruments).

**Statistics**. Most data are presented as dot plots with means ± SD or box and whiskers plots, where the upper and lower margins of the box define the 75th and 25th percentiles, respectively, the internal line defines the median, and the whiskers show the total range. Bar graphs showing means ± SEM were used for data normalized to a control group (e.g., qRT-PCR results, where the control group was assigned a value of one) and for Fig. 3, showing averages of at least three independent cell culture experiments. Where appropriate (for $n > 7$), the data were tested for normality using the D'Agostino-Pearson omnibus normality test, and for equal variance using an $F$ test (to compare the variances of two groups), or the Browne and Forsythe test (for three or more groups). For comparison of two groups, $P$ values refer to unpaired, two-tailed Student's $t$-test. For multiple comparisons, $P$ values refer to either one-way ANOVA followed by Sidak's multiple comparison test, or to two-way ANOVA with Holm–Sidak's multiple comparison test (e.g., when determining how the genotype of the mice affected response to drug treatment or TAC). $P < 0.05$ was considered statistically significant. When the assumptions of normal distribution and/or equal variances were not met, a non-parametric test was chosen (Mann–Whitney test for two groups and Kruskal–Wallis test for three or more groups). Data analysis was performed using GraphPad Prism 7 software. The tests employed are described in the figure legends.

**Reporting Summary**. Further information on research design is available in the Nature Research Reporting Summary linked to this article.

## Data availability

All relevant data supporting the findings of this study are available within the paper and its supplementary information files. The source data underlying all graphs and blots in Figs. 1–5 and Supplementary Figs. 1–8 are provided as a Source Data File. All data are available from the corresponding author upon reasonable request.

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

## Acknowledgements
This work was supported by the National Institutes of Health grants R01-HL132141 (to R.B.P.), U01 NS58030 (to G.R.B.), and P30-NS047101 (UCSD Microscopy Shared Facility). G.K.S. was supported by the Deutsche Forschungsgemeinschaft.

## Author contributions
Study design: G.K.S., D.M.M., K.L.P., G.R.B. and R.B.P. Study conduct: G.K.S., H.K., D.E.C., N.D.D., Y.G., S.L., S.Z., W.W., J.M.S., Q.Z. and A.M. Data collection: G.K.S., H.K., N.D.D., Y.G. and Q.Z. Data analysis: G.K.S., H.K. and R.B.P. Data interpretation: G.K.S., H.K., H.H.P., D.M.M. K.L.P., G.R.B. and R.B.P. Drafting manuscript: G.K.S., G.R.B. and R.B.P. G.K.S., H.K. and R.B.P. take responsibility for integrity of data analysis.

## Additional information

**Competing interests:** The authors declare no competing interests.

