## [Peer Review File · Nature Communications]

Reviewers' comments:

Reviewer #1 (Remarks to the Author):

This is interesting work with a description of a relatively clear phenotype in a transgenic mouse that expresses a mutation relevant to human disease, that results in expression of over-active PKG.

Whilst there is a lot of data, and I am convinced by the observations, I am not so convinced by the overall mechanism that is proposed. It has been very common for cardiovascular and other diseases to be attributed to oxidative stress, but overall the role of oxidants in cardiovascular pathogenesis has probably been overstated. Certainly, there are no large-scale antioxidant therapies that work in humans, and often such interventions have shown indications of harm. Once the tissue is in a different state (as with aneurism) markers of oxidative stress are likely to be present. But are these oxidative modifications causal or a consequence of the disease? As antioxidants block the adverse disease progression in this study, the authors claim that oxidants (derived from Nox4) are causal. But, I am mindful that vast numbers of other pre-clinical studies have made similar claims that oxidants mediate disease / dysfunction observed in experimental models of many different diseases. Given antioxidant interventions have been so widely claimed to be protective, but at the same time have so clearly failed to translate to human therapies, I think that at the current state of play we need evidence beyond antioxidants limiting indices of oxidations and protecting against disease progression. This is especially important given the harm antioxidants have caused in the setting of disease as evidenced by large scale clinical trials.

If you simply over express Nox4 – do you see the same phenotype as the PKG mutant? Or what does the literature on such mice suggest? There is at least some evidence for Nox4 providing cardiovascular protection. <http://www.pnas.org/content/107/42/18121.long>

Are Nox inhibitors protective?

Are there particular targets of Nox4 that are especially important in mediating the dysfunction.

Are the PKG activities assay completely selective for PKG. PKA may be taken care of by the assay design, but other kinase could have a role. Is the activity measured completely inhibited by a selective PKG inhibitor?

Figure 3b – the DNPH - / + labelling may be the wrong way around?

Please avoid presenting assay data with normalisation, as currently use in Figure 2o and p. Please use numbers with real units, not arbitrary units.

What was the impact of the antioxidant interventions on PKG activity? This is especially important given the redox regulation of this kinase.

Are the responses of the blood vessels *ex vivo* different than wild types? For example, to nitric oxide, acetylcholine, PKG activators, or constrictor agonists. It is perhaps important to understand why the blood pressure is not different, which may have been expected due to activation of PKG by the mutation and the well-established role of this kinase in blood pressure. Were there compensatory changes in heart rate, or activity of the mice - which the DSI system should provide data on. Is there a trend for mutant mouse to have lower blood pressure, and what are its day and night diastolic / systolic / MAP / pulse pressure data? If these various hemodynamic measures are assessed carefully, are there differences between genotypes? Coming back to why there is no difference in blood pressure, what adaptive change might account for this? Is the blood volume, or renal function altered?

Reviewer #2 (Remarks to the Author):

In this manuscript the authors studied the mechanisms how an activating mutation of PGK promotes development of aortic aneurysms. The report that a heterozygous knockin mouse exhibits increased aneurysm development and increased aneurysm-associated mortality. Treatment with two different antioxidants attenuated this effect. Mechanistically, the authors observed increased oxidative stress and protein oxidation in response to the activating mutation.

This is an interesting study but some molecular mechanisms remain insufficiently addressed. This reviewer has the following specific comments:

- 1) The authors fail to provide evidence that it is truly that activating mutation of PGK and not some other problems of the knock-in strategy which mediates the effect. Some of the *ex vivo* experiments should be repeated in the presence of a PGK inhibitor to demonstrate that this compound indeed blocks the effect of the knockin.
- 2) As evident from Fig 4j, the pathway from PKG1 activation to Nox4 and TGF β 1 is complex and not really clear. The true molecular mechanism of action of the PKG mutation has to be identified. What is the mechanism of Nox4 induction? Is this through TGF β 1 or through some other

mechanisms? What is the mechanism of TGF beta increase? Experiments in cultured cells or organ explants are needed to reveal the underlying mechanism.

3) PKG is downstream of cGMP, which is its main activator. Is there any evidence that drugs which promote cGMP signaling (like PDE-V inhibitors) have a similar effect? What happens to gene expression or ROS production in this study if cells were treated with PKG activators or as an alternative with PDE inhibitors in the presence of low doses of ANP or NO?

4) The B12-based antioxidant is an interesting compound. Does it have any remnant effects on the B12 system?

Reviewer #3 (Remarks to the Author):

This manuscript reports thoracic aortic pathology in Prkg1 R177Q/+ mice (4-6 months versus 12 months) and thoracic aortic pathology induced by transverse aortic constriction procedure (TAC). The manuscript presents a large body of data, and the general theme is interesting. However, there are major concerns regarding experimental design, measurements, and data analyses.

Comments:

1. Statistical analysis is one major concern of this manuscript. The authors used either Student's t-test or one way ANOVA with Bonferroni post hoc test without testing data normality and equal variance. In addition, many data should be analyzed using two factor analysis such as two way ANOVA if data pass normality and equal variance tests. The "reporting Summary" claims that the manuscript contains a "description of any assumptions...", but this is not present. These include Figure 2j, l, m, o, p, and q, Fig. 3b, e, d, e, and f, Fig4c, d, e, f, g, h, l, and many data shown in Supplemental figures. In the absence of knowledge that appropriate statistical tests were applied, it is difficult to determine whether the data supports the conclusions.

2. Given the strong sex differences in aorta diseases, it is not appropriate to report groups of mixed sex in the absence of data demonstrating that sex does not impact the data. Also, it is inappropriate not to state the number of male and females in each group.

3. The major conclusion of this manuscript is "aortic degeneration due to PKG activation is prevented by an antioxidant vitamin B12 analog cobinamide". "Aortic degeneration" is an ambiguous term, which was not defined clearly in this manuscript. The major data related to aortic pathology in this manuscript include aortic dilation, elastin fiber breaks, and collagen content in the thoracic aorta. The authors should make clear the focus is "thoracic aortic pathology".

4. Figure 1d and 1e show ultrasound data. Since A - D measurements are all above LSA, "descending aorta" was not included. Therefore, the data for "descending aorta" were not the

descending aorta. Three parts are shown on the ultrasound images: aortic root, ascending aorta, and aortic arch. Aortic valve can not be clearly defined and also should not be part of aortic dilation measurements. Ultrasound measurements have not been consistent: Figure 3d only showed aortic valve (should be aortic root), ascending aorta, and transverse aorta (should be “aortic arch”), Figure 4C only presented ascending aorta and Suppl. Fig. 6d only showed aortic valve (“aortic root”) diameter, but Suppl. Fig. 2e showed 4 locations. The presentation formats are also different with some showing individual plots, Fig.3d shows box, and Fig. 4 and Suppl. Fig. 6d show changes between groups.

5. As shown by ultrasound, differences of aortic diameters between the two genotypes in different locations are modest. This is concerning given the measurement were derived from a technique that is user skill and experience-dependent. Given this concern, there should be documentation on how consistent measurements were obtained. For example, the changes appear to be within the range (about 0.2 mm) that the aorta changes in this region in systole versus diastole. Were measurements taken at a specific phase of the cycle?.

6. The authors state that histology was performed in the ascending aortic region, but many images shown in this manuscript do not appear to be from the ascending region. For example, images shown in Fig.3f, it appears that images for WT control and Cbi are from the ascending region, but the images for RQ/+ were from another region. It also appears that the magnifications are different. The authors should check and confirm that all images were from same location of the aorta. Since most experience observers would use the number of elastin layers to determine location of tissues sections, an image claiming to be from region that does not have the commonly accepted number of elastin fibers should come with an explanation of the variance.

7. Quality of images for TUNEL, 8-OH-Deoxyguanosine, and DHE staining are poor. No appropriate negative controls were provided.

8. Figure 4b shows the most interesting data in this manuscript: TAC-induced aortic rupture in RQ/+ mice, but not in WT mice. However, it is hard to see whether this is “aortic rupture” from the two images shown in Suppl. Fig. 6C - ex vivo images of aortas would provide more meaningful information.

9. Figure 4b: The authors did not provide information whether RQ/+ versus RQ/+ NAC groups were different, but stated in the abstract and the results that “NAC ameliorated TAC-induced pathology and mortality”, which are not consistent with the data shown. Figure 4c and Suppl Fig. 6d: It would be important to compare aortic diameters among and between groups, not only within each group pre and post surgery.

10. Suppl. Fig. 6e: RA/+ image: Are the author coincident that this is tear rather than a branch of the coronary ostia. This low magnification single image does not provide meaningful information to understand whether a tear is present.

11. In the method, the authors also stated that for TAC experiments, mice were backcrossed for at least 8 generations into C57BL/6N background, but did not provide whether other experiments used mice with the same background.

Re: Revision of Manuscript # NCOMMS-18-34576-T**Comments by the reviewers are in italics, followed by our responses in regular font .****Reviewer #1:**

This is interesting work with a description of a relatively clear phenotype in a transgenic mouse that expresses a mutation relevant to human disease, that results in expression of over-active PKG.

Whilst there is a lot of data, and I am convinced by the observations, I am not so convinced by the overall mechanism that is proposed. It has been very common for cardiovascular and other diseases to be attributed to oxidative stress, but overall the role of oxidants in cardiovascular pathogenesis has probably been overstated. Certainly, there are no large-scale antioxidant therapies that work in humans, and often such interventions have shown indications of harm. Once the tissue is in a different state (as with aneurism) markers of oxidative are likely to be present. But are these oxidative modifications causal or a consequence of the disease? As antioxidants block the adverse disease progression in this study, the authors claim that oxidants (derived from Nox4) are causal. But, I am mindful that vast numbers of other pre-clinical studies have made similar claims that oxidants mediate disease / dysfunction observed in experimental models of many different diseases. Given antioxidant interventions have been so widely claimed to be protective, but at the same time have so clearly failed to translate to human therapies, I think that at the current state of play we need evidence beyond antioxidants limiting indices of oxidations and protecting against disease progression. This is especially important given the harm antioxidants have caused in the setting of disease as evidenced by large scale clinical trials.

1. [NOX4 over-expression.] *If you simply over express Nox4 – do you see the same phenotype as the PKG mutant? Or what does the literature on such mice suggest? There is at least some evidence for Nox4 providing cardiovascular protection. <http://www.pnas.org/content/107/42/18121.long>* Whether NOX4 exerts adverse or protective actions in the cardiovascular system is the subject of an ongoing debate, with discrepant results under seemingly similar experimental settings. While basal NOX4-derived H₂O₂ may have vascular-protective effects, increased NOX4 expression can exert both adverse and protective actions in cardiovascular disease models.¹⁻⁶ For example, in the paper referred to by the reviewer,¹ the authors found that young mice with cardiomyocyte-specific over-expression of NOX4 (NOX4-CM-Tg) are protected from cardiac fibrosis and dilation after pressure overload, because of increased angiogenesis from increased *Hif1α* and *Vegfa* expression. However, other workers have found worse cardiac fibrosis and dilation after pressure overload in NOX4-CM-Tg mice (in a slightly different background strain), associated with increased apoptosis and oxidative stress markers.⁶ In the vasculature, NOX4 can be either protective (e.g., anti-atherogenic) or deleterious (e.g., in pulmonary hypertension),^{4,5} but global deletion of NOX4 appears to be deleterious, because physiological levels of NOX4 activity are required for regulation of *Hif-1α/Vegf* and other factors.⁷ Moreover, aortic SMCs isolated from global *Nox4*^{-/-} mice show a de-differentiated, hyper-proliferative phenotype with up-regulation of *Nox1* and increased ROS production.² These data are now discussed on page 10 and show that changes in NOX4 expression can have complex effects *in vivo*. Thus, rather than overexpressing NOX4 *in vivo*, where results could be confusing, we expressed NOX4 in human aortic SMCs. We found

increased H₂O₂ production, an increase in the oxidative stress marker 8-hydroxy-deoxyguanosine (8-OH-dGuo), and increased apoptosis and decreased proliferation compared to control cells (Fig. 3g,k,m and o). Thus, NOX4 overexpression in human aortic SMCs mimicked the effects of mutant PKG1^{RQ}.

2. Are Nox inhibitors protective?

As described above, the role of NOX4 *in vivo* is complicated, and treating mice with the NOX1/4 inhibitor GKT137831⁸ caused a 5-fold up-regulation of NOX1 mRNA in the aorta, resulting in little or no reduction in ROS production.⁹ Thus, we tested the effects of NOX1/4 inhibition in human aortic SMCs expressing mutant PKG1^{RQ}: GKT137831 inhibited excess H₂O₂ generation, confirming previous experiments with NOX4 shRNA knock-down (Fig. 3g); the drug also reduced apoptosis and DNA oxidation, and partly restored proliferation (Fig. 3j,l,n). These results confirm that NOX4-derived H₂O₂ contributes to the increased oxidative stress, apoptosis, and impaired proliferation seen in SMCs expressing the mutant PKG1^{RQ}.

3. Are there particular targets of Nox4 that are especially important in mediating the dysfunction?

Transgenic mice with targeted overexpression of *Nox4* in cardiac myocytes or glomerular podocytes have increased *Hif-1α* and *Tgfb1* mRNA and protein expression in the targeted cells, while *Nox4* knockout mice have reduced levels of *Hif-1α*, suggesting that *Hif-1α* and *Tgfb1* are NOX4 targets in multiple cell types.^{1,7,10} *Hif-1α* regulates *Vegfa* expression, while *Tgfb1* regulates *Ctgf*, and we found increased mRNA expression of *Hif-1α*, *Vegfa*, *Tgfb1*, and *Ctgf* in aortas of *Prkg1*^{RQ/+} mice compared to wild type litter mates, consistent with increased *Nox4* expression in the mutant aortas (Fig. 4g). Importantly, cobinamide (Cbi) treatment reduced all five transcripts in the mutant mice, with little effect in wild type mice (Fig. 4g). Since dysregulation of the *Hif-1α/Vegfa*, and *Tgfb1/Ctgf* pathways has been linked to aortic aneurysm progression,¹¹⁻¹⁴ our data suggest that these *Nox4* target genes likely contribute to aortic pathology in PKG1^{RQ/+} mice. The *Nox4* promoter itself is redox-sensitive,¹⁵ and reducing oxidative stress with Cbi can largely correct the excessive expression of *Nox4*, *Hif-1α/Vegfa*, and *Tgfb1/Ctgf* in the mutant mice. These results are now described on page 8 and discussed on page 9.

4. Are the PKG activities assay completely selective for PKG? PKA may be taken care of by the assay design, but other kinases could have a role. Is the activity measured completely inhibited by a selective PKG inhibitor?

Interference by PKA activity is indeed prevented by the PKA-specific protein kinase inhibitor (PKI) included in the assay. Peptide inhibitors of the DT2 series are the only available PKG-selective inhibitors that inhibit basal kinase activity (in the absence of cGMP).¹⁶ Cyclic nucleotide analogs have partial agonist activity and KT5823 is ineffective.¹⁷ DT2 at 10 μM inhibited purified wild type PKG1α and mutant PKG1α^{RQ} by ~80%, and in aortic extracts from wild type and PKG1^{RQ/+} mice, DT2 inhibited basal and cGMP-stimulated activity to a similar extent (Suppl. Fig. 1g and Fig. 1a). Thus, the increased basal PKG activity measured in mutant aortas does not reflect activity of other protein kinases.

5. Figure 3b – the DNPH - / + labelling may be the wrong way around?

Yes, we apologize, and have corrected the mistake (new Fig. 4b).

6. Please avoid presenting assay data with normalization, as currently used in Figure 2o and p. Please use numbers with real units, not arbitrary units.

Previous Fig. 2o (new Fig. 3c): mRNA quantification by qRT-PCR using the comparative ΔΔCT method¹⁸ requires normalization to the control condition (which is the mean of ΔCT values obtained in wild type mice). As is done by most other investigators, we did not perform standard curves for each transcript

to calculate absolute copy numbers, because preparation of standard curves carries a high risk of contamination.

Previous Fig. 2p (new Fig. 3f): H₂O₂ production is now expressed as Amplex Red fluorescence units per μg protein.

7. What was the impact of the antioxidant interventions on PKG activity? *This is especially important given the redox regulation of this kinase.*

Adding Cbi (10 μM to 1 mM) to purified PKG1 α^{RQ} had no effect on PKG activity (Suppl. Fig. 7a). We also measured PKG activity in aortic extracts from wild type and PKG1 $\text{RQ}/+$ mice treated for one month with 1 mM Cbi or vehicle, and found no effect of Cbi on basal or cGMP-stimulated PKG activity (Suppl. Fig. 7b). In terms of potential redox regulation of PKG, we assessed Cys⁴³ oxidation of PKG1 α in aortic extracts by a gel shift assay under non-reducing conditions.¹⁹ We found a small percentage (~10 %) of cross-linked PKG1 α dimer (representing Cys⁴³-oxidized enzyme), but found no difference in the amount of “oxidized” PKG1 between wild type and heterozygous PKG1 $\text{RQ}/+$ aortas, or between untreated and Cbi-treated mice (Suppl. Fig. 7d). As a positive control, we treated human SMCs with high concentrations of H₂O₂ (i.e., 100-200 μM); this doubled the amount of cross-linked, oxidized PKG1 α , without increasing PKG activity, as determined by VASP phosphorylation (Suppl. Fig. 7e). These data are consistent with our previous finding that PKG activity is not directly regulated by oxidation-induced disulfide formation at Cys⁴³; the significance of PKG1 redox regulation *in vivo* is controversial.^{19,20}

8. Are the responses of the blood vessels ex vivo different than wild types? *For example, to nitric oxide, acetylcholine, PKG activators, or constrictor agonists. It is perhaps important to understand why the blood pressure is not different, which may have been expected due to activation of PKG by the mutation and the well-established role of this kinase in blood pressure. Were there compensatory changes in heart rate, or activity of the mice - which the DSI system should provide data on. Is there a trend for mutant mouse to have lower blood pressure, and what are its day and night diastolic / systolic / MAP / pulse pressure data? If these various hemodynamic measures are assessed carefully, are there differences between genotypes? Coming back to why there is no difference in blood pressure, what adaptive change might account for this? Is the blood volume, or renal function altered?*

We re-analyzed telemetry data for wild type and PKG1 $\text{RQ}/+$ mice, averaging mean arterial pressure (MAP) and pulse pressure separately during hours of activity (7 pm to 7 am) *versus* rest (7 am to 7 pm). During both time periods, the MAP of heterozygous PKG1 $\text{RQ}/+$ mice was ~ 10 mmHg lower than that of wild type litter mates (Fig. 1d), while no difference in pulse pressure or heart rate was observed and circadian variations in blood pressure were similar (Suppl. Fig. 2a-c; described on page 4). No difference in renal function, based on serum BUN and creatinine, were observed between the two genotypes (Suppl. Table 1). These data of small blood pressure changes in the PKG1 $\text{RQ}/+$ mice are consistent with data from global PKG1 knockout mice, which are only transiently and mildly hypertensive.²¹

We performed isometric tension measurements of aortic rings from wild type and PKG1 $\text{RQ}/+$ mice. The prostaglandin F₂ α concentration required to pre-contract rings to 0.15 g was similar in aortas from wild type and mutant mice (described on page 4). However, relaxation of rings in response to 8-CPT-cGMP or to acetylcholine—which is largely NO-mediated²²—was reduced in the PKG1 $\text{RQ}/+$ mice compared to wild type mice (Suppl. Fig. 2d,e). These data are consistent with high basal, cGMP-independent PKG activity in the mutant aortas, resulting in smaller cGMP-induced increases in enzyme activity compared to wild type aortas (Suppl. Fig. 2f). Reduced NO bioavailability due to oxidative stress in the mutant mice could contribute to decreased acetylcholine-induced aortic relaxation, but this requires further investigation (discussed on page 4). More work is also required to examine resistance vessels in the mutant mice, but this is beyond the scope of the present manuscript.

Reviewer #2 (Remarks to the Author):

In this manuscript the authors studied the mechanisms how an activating mutation of PGK promotes development of aortic aneurysms. The report that a heterozygous knockin mouse exhibits increased aneurysm development and increased aneurysm-associated mortality. Treatment with two different antioxidants attenuated this effect. Mechanistically, the authors observed increased oxidative stress and protein oxidation in response to the activating mutation.

This is an interesting study but some molecular mechanisms remain insufficiently addressed. This reviewer has the following specific comments:

1) *The authors fail to provide evidence that it is truly that activating mutation of PGK and not some **other problems of the knock-in strategy which mediates the effect. Some of the ex vivo experiments should be repeated in the presence of a PGK inhibitor to demonstrate that this compound indeed blocks the effect of the knockin.***

The PKG1^{RQ} mutant mice were generated using a traditional knock-in strategy (described in Suppl. Fig. 1a-e), which is less prone to off-target effects than the CRISPR/Cas9 system. Moreover, the knock-in mice were back-crossed (please see comment 11 of reviewer #3, below), which further decreased the chances of additional genetic lesions. A major argument against off-target effects is that the main phenotypic changes in aortas from the mutant mice were reproduced in two different strains (129S and C57Bl/6), as well as in human aortic SMCs expressing the mutant PKG1^{RQ} enzyme; these include changes in gene expression and oxidative stress markers (Figs. 1, 2 and 4 for mice in the 129S background, Fig. 5 for mice in the C57Bl/6 background, and Fig. 3 for human aortic SMCs).

We treated human aortic SMCs expressing PKG1^{RQ} with DT2, the specific PKG inhibitor which effectively blocked basal activity of the mutant PKG1^{RQ} *in vitro* (please see comment 4 of reviewer #1, above). DT2 treatment reduced apoptosis and oxidative stress markers, and restored proliferation in PKG1^{RQ} expressing SMCs, indicating that these PKG1^{RQ} effects indeed required PKG activity (Fig. 3j,l,n, described on page 6 and 7).

2) *As evident from Fig 4j, the pathway from PKG1 activation to Nox4 and TGF- β 1 is complex and not really clear. The true molecular mechanism of action of the PKG mutation has to be identified. **What is the mechanism of Nox4 induction? Is this through TGF beta1 or through some other mechanisms? What is the mechanism of TGF beta increase? Experiments in cultured cells or organ explants are needed to unveil the underlying mechanism.***

To study mechanism(s) whereby PKG1^{RQ} increased *Nox4* expression, we constructed a luciferase reporter under control of the mouse *Nox4* promoter. Treating cells with TGF- β increased luciferase activity almost three-fold, consistent with the effect of TGF β on the human *NOX4* promoter;¹⁵ this stimulation was blocked in the presence of SB-505124, a well-characterized inhibitor of the TGF- β receptor-1.²³ PKG1^{RQ} also increased luciferase expression almost three-fold; however, this was not affected by the TGF- β receptor inhibitor, suggesting that PKG1^{RQ} stimulation of *Nox4* was TGF- β independent (Suppl. Fig. 4g).

PKG1^{RQ} activates N-terminal Jun kinase (JNK) in SMCs (Figs. 2h and 3h), which can phosphorylate and activate the AP-1 subunit c-Jun; the proximal *Nox4* promoter contains two canonical AP-1 binding sites (at -1352 and -1337 with respect to the translational start site, conserved in the human promoter). PKG1^{RQ} enhanced the stimulatory effect of c-Jun on the promoter in a JNK-dependent manner, suggesting that PKG1^{RQ} stimulation of *Nox4* is mediated by JNK/cJun (Suppl. Fig. 4h). Similarly, 7-ketocholesterol increases *NOX4* transcription in human SMCs via activation of JNK/c-Jun.²⁴ These results are described on page 6/7.

The diagram describing pathways controlled by PKG1^{RQ} (previous Fig. 4j) was revised to reflect the new findings that PKG likely activated Nox4 via JNK, independently of TGF- β (now Fig. 5j). The diagram additionally shows that TGF- β is both upstream and downstream of Nox4, supported by our findings as well as by reports of others (discussed on pages 8/9).

3) PKG is down-stream of cGMP, which is its main activator. **Is there any evidence that drugs which promote cGMP signaling (like PDE-V inhibitors) have a similar effect?** What happens to gene expression or ROS production in this study if cells were treated with PKG activators or as an alternative with PDE inhibitors in the presence of low doses of ANP or NO?

We treated human aortic SMCs with low doses of the NO donor DETA-NONOate, the PDE-5 inhibitor sildenafil, and a combination of the two, and found at least additive activation of endogenous PKG1 by the combination (Suppl. Fig. 5f). The two drugs activated JNK and inhibited SMC proliferation, thus mimicking important effects of PKG1^{RQ} (Suppl. Fig. 5d-f).

In vivo experiments are ongoing to test effects of long-term sildenafil administration on thoracic aortic pathology in wild type mice and mice with different genetic predispositions for TAAD.

4) **The B12-based antioxidant is an interesting compound. Does it have any remnant effects on the B12 system?**

Cobinamide is the penultimate precursor in vitamin B₁₂ biosynthesis, and low concentrations are found in human serum. Cbi does not bind to intrinsic factor, the protein required for B₁₂ intestinal absorption, nor to the main B₁₂ binding proteins in serum; it, therefore would not be expected to compete for B₁₂ absorption or transport. Mice treated with Cbi for 6 months (1 mM in the drinking water, Fig. 4) had normal red blood cell counts and mean corpuscular volumes, sensitive indicators that the mice were not functionally B₁₂ deficient (Suppl. Table 2). In addition, they had normal serum homocysteine and methyl-malonate concentrations, indicating that Cbi had no measurable effects on the activities of methionine synthase and methylmalonyl-CoA mutase, the two mammalian enzymes requiring vitamin B₁₂ as a co-factor (Suppl. Table 1).

Reviewer #3 (Remarks to the Author):

This manuscript reports thoracic aortic pathology in Prkg1 R177Q/+ mice (4-6 months versus 12 months) and thoracic aortic pathology-induced by transverse aortic constriction procedure (TAC). The manuscript presents a large body of data, and the general theme is interesting. However, there are major concerns regarding experimental design, measurements, and data analyses.

Comments:

1. **Statistical analysis is one major concern of this manuscript.** *The authors used either Student's t-test or one way ANOVA with Bonferroni post hoc test without testing data normality and equal variance. In addition, many data should be analyzed using two factor analysis such as two way ANOVA if data pass normality and equal variance tests. The "reporting Summary" claims that the manuscript contains a "description of any assumptions...", but this is not present. These include Figure 2j, l, m, o, p, and q, Fig. 3b, e, d, e, and f, Fig4c, d, e, f, g, h, l, and many data shown in Supplemental figures. In the absence of knowledge that appropriate statistical test were applied, it is difficult to determine whether the data supports the conclusions.*

Where appropriate (n>7), we have tested for data normality using the D'Agostino-Pearson omnibus normality test, and for equal variance using an F test to compare the variances of two groups and the Browne and Forsythe test for 3 or more groups. When these assumptions were not met, a non-

parametric test was chosen, Mann-Whitney test for two groups or Kruskal-Wallis test for three or more groups. The assumption of normal distribution and equal variances prior to performing parametric tests is now described in the Statistics Section (page 5/6 of Supplemental Methods).

We apologize for not having employed a two-factor ANOVA when appropriate: the data in Fig. 1a, Fig. 3 a, d-g, j-o (Fig. 3 was previously part of Fig. 2), Fig. 4 a-g, Fig. 5c-l, Suppl. Fig. 4 a,d-h, Suppl. Fig. 5a-d and f, Suppl. Fig. 6b and c, and Suppl. Fig. 7b have now been analyzed using a two-way ANOVA with Holm-Sidak's multiple comparison test. The tests employed are described in the figure legends.

2. Given the strong sex differences in aorta diseases, it is not appropriate to report groups of mixed sex in the absence of data demonstrating that sex does not impact the data. Also, it is inappropriate not to state the number of male and females in each group.

We analyzed aortic diameters of aging male and female mice separately, by pooling 12-month-old males and females from two different studies. The aortic root, ascending aorta, and aortic arch diameters are significantly different between wild type and PKG1^{RQ/+} mutant mice, with no gender-specific difference (by two-way ANOVA, data shown in Suppl. Fig. 2g and h; for aortic root, the data in mutant mice showed increased variance, but the differences between wild type and mutant mice were confirmed by Kruskal-Wallis test). The number of male and female mice in each group is now stated in all figure legends. TAC experiments were performed only in male mice for homogeneity of results.

3. The major conclusion of this manuscript is "aortic degeneration due to PKG activation is prevented by an antioxidant vitamin B12 analog cobinamide". "Aortic degeneration" is an ambiguous term, which was not defined clearly in this manuscript. The major data related to aortic pathology in this manuscript include aortic dilation, elastin fiber breaks, and collagen content in the thoracic aorta. The authors should make clear the focus is "thoracic aortic pathology".

We now use the term "thoracic aortic pathology" instead of "aortic degeneration" throughout the manuscript (although we left "aortic degeneration" for brevity in the title). We now define this term on page 2 as encompassing aortic dilation, progressive elastin fiber fragmentation, SMC loss, and collagen accumulation.

4. Figure 1d and 1e show ultrasound data. Since A - D measurements are all above LSA, "descending aorta" was not included. Therefore, the data for "descending aorta" were not the descending aorta. Three parts are shown on the ultrasound images: aortic root, ascending aorta, and aortic arch. Aortic valve cannot be clearly defined and also should not be part of aortic dilation measurements. Ultrasound measurements have not been consistent: Figure 3d only showed aortic valve (should be aortic root), ascending aorta, and transverse aorta (should be "aortic arch"), Figure 4C only presented ascending aorta and Suppl. Fig. 6d only showed aortic valve ("aortic root") diameter, but Suppl. Fig. 2e showed 4 locations. The presentation formats are also different with some showing individual plots, Fig.3d shows box, and Fig. 4 and Suppl. Fig. 6d show changes between groups.

The description of "aortic valve" has been changed to "aortic root", and "transverse aorta" to "aortic arch;" "ascending aorta" remained the same, and we eliminated data previously labeled as "descending aorta" (Fig. 1e shows the three measurements for aging mice, Fig. 5a and Suppl. Fig. 8a show mice before and after TAC surgery). In Fig. 1 and Suppl. Fig. 2, we show dot blots of individual wild type and heterozygous PKG^{RQ/+} mice at 12 months of age, because only two groups are compared. In Fig. 4 and 5, we chose less noisy box plots with whiskers showing median and interquartile range and the total range to compare four groups of mice: Wt and RQ/+, treated and untreated. ECHO data are presented for aortic root, ascending aorta, and aortic arch, except Fig. 5 shows only ascending aorta because of space constraints, with aortic root shown in Suppl. Fig. 8d.

5. As shown by ultrasound, differences of aortic diameters between the two genotypes in different locations are modest. This is concerning given the measurement were derived from a technique that is user skill and experience-dependent. Given this concern, there should be **documentation on how consistent measurements were obtained**. For example, the changes appear to be within the range (about 0.2 mm) that the aorta changes in this region in systole versus diastole. **Were measurements taken at a specific phase of the cycle?**

All ultrasound measurements were performed by a single, experienced observer who was blinded to the genotype and mouse treatment. Measurements were taken at end-diastole, using the ECG for timing, as shown by an arrow head below the ultrasound images and ECGs in Figs. 1e and 5a.

We analyzed inter- and intra-observer variability using Bland Altman plots (plots are shown and described in the Supplemental Methods Section, page 2). For measurements of the ascending aorta, the average of the differences between two observers was -0.061 mm (with 95% limits of agreement between -0.18 and 0.06 mm), and the average of differences between two observations by the same observer were 0.024 mm (with 95% limits of agreement between -0.12 and 0.17 mm). Thus, changes in aortic diameter within the range of 0.1-0.2 mm could be measured accurately, and the differences observed between 12 month-old PKG1^{RQ/+} mice and wild type littermates are similar in magnitude to those reported in other mouse models of aortic aneurysms.²⁵

6. The authors state that histology was performed in the ascending aortic region, **but many images shown in this manuscript do not appear to be from the ascending region**. For example, images shown in Fig.3f, it appears that images for WT control and Cbi are from the ascending region, but the images for RQ/+ were from another region. It also appears that the magnifications are different. The authors should check and confirm that all images were from same location of the aorta. Since most experience observers would use the number of elastin layers to determine location of tissues sections, an image claiming to be from **region that does not have the commonly accepted number of elastin fibers should come with an explanation of the variance**.

We have recut and re-stained some of the tissue blocks and carefully checked that all images shown are from the ascending region of the aorta, taken with the same magnification. Sections in the previous Figs. 3f (new Fig. 4f) and 4e (new Fig. 5e) were replaced. In untreated PKG1^{RQ/+} mice, we sometimes observed loss of adventitia and outer media layers, most likely due to tissue shear injury during dissection of these more fragile aortas, but this was not a consistent finding.

7. **Quality of images for TUNEL, 8-OH-Deoxyguanosine, and DHE staining are poor. No appropriate negative controls were provided.**

We apologize that the previously provided pdf version of the figures was not of high resolution and did not include negative controls. We have replaced most of the TUNEL and 8-OH-deoxyguanosine stained sections with new images showing better contrast and resolution, and we have included negative controls for all immunohistochemical stains (Fig. 1i, Fig. 2d, Fig. 4a, Suppl. Fig. 6a). Images of DHE-stained aortic sections have been replaced with brighter, higher resolution images (Suppl. Fig. 4). We have carefully reviewed all histochemical and immunohistochemical stains with colleagues, who found them of a quality consistent with the quality of histological images seen in publications in Nature Journals.

8. **Figure 4b shows the most interesting data in this manuscript: TAC-induced aortic rupture in RQ/+ mice, but not in WT mice. However, it is hard to see whether this is “aortic rupture” from the two images shown in Suppl. Fig. 6C - ex vivo images of aortas would provide more meaningful information.**

Unfortunately, necropsies were performed several hours after the death of the animals, and dissection of the aortas for ex vivo imaging was not possible at that time. Thus, we have only photographic documentation of mice dying from hemothorax and hemopericardium. The pictures shown in Suppl Fig. 8c have been enlarged.

9. Figure 4b: *The authors did not provide information **whether RQ/+ versus RQ/+ NAC groups were different**, but stated in the abstract and the results that “NAC ameliorated TAC-induced pathology and mortality”, which are not consistent with the data shown. Figure 4c and Suppl Fig. 6d: It would be important to **compare aortic diameters among and between groups**, not only within each group pre and post surgery.*

The survival curve (previous Fig. 4b, new Fig. 5b) shows that significantly more PKG^{RQ/+} mice died after TAC compared to wild type mice; numerically, less NAC-treated PKG^{RQ/+} mice died compared to untreated mutant mice, but the difference did not reach statistical significance. This is now clearly stated on page 8/9, and the abstract says that “NAC ameliorated TAC-induced pathology,” referring to the histological improvements seen in NAC-treated PKG^{RQ/+} mice (Fig. 5d-i).

We now compare aortic diameters between NAC-treated and control, wild type and PKG^{RQ/+} mice, before and after TAC (8 groups total). By two-way ANOVA, the only significant differences found were the comparison before *versus* after TAC. These data with the more extensive comparisons are shown in Fig. 5c (previously 4c) and Suppl. Fig. 8d (previously 6d) and are discussed on page 8/9.

10. *Suppl. Fig. 6e: RA/+ image: Are the author coincident **that this is tear rather than a branch of the coronary ostia**. This low magnification single image does not provide meaningful information to understand whether a tear is present.*

Additional cuts of the mid-ascending aorta could not confirm the presence of a tear in multiple sections; therefore, the previous Suppl. Fig. 6e was removed.

11. *In the methods, the authors also stated that for TAC experiments, mice were backcrossed for at least 8 generations into C57BL/6N background, but **did not provide whether other experiments used mice with the same background**.*

As stated in Supplemental Methods (page 1), the knock-in mice were generated in a 129S1/4 background and crossed for three generations into the 129S1/SvImJ background (JAX mouse stock #002448). These 129S1 mice were used in all experiments, except for the TAC experiments shown in Fig. 5 and Suppl. Fig. 8. For the latter experiments, mice were backcrossed into a C57BL6/6NHsd background for at least eight generations, because cardiac outcomes after TAC are well characterized in male mice of this background and these mice show no evidence of heart failure 14 d after TAC.²⁶

REFERENCES (in Response to Reviewers’ Comments)

1. Zhang, M., *et al.* NADPH oxidase-4 mediates protection against chronic load-induced stress in mouse hearts by enhancing angiogenesis. *Proc Natl Acad Sci U S A* **107**, 18121-18126 (2010).
2. Di Marco, E., *et al.* NOX4-derived reactive oxygen species limit fibrosis and inhibit proliferation of vascular smooth muscle cells in diabetic atherosclerosis. *Free Radic Biol Med* **97**, 556-567 (2016).
3. Schroder, K., *et al.* Nox4 is a protective reactive oxygen species generating vascular NADPH oxidase. *Circ Res* **110**, 1217-1225 (2012).
4. Fulton, D.J. & Barman, S.A. Clarity on the Isoform-Specific Roles of NADPH Oxidases and NADPH Oxidase-4 in Atherosclerosis. *Arterioscler Thromb Vasc Biol* **36**, 579-581 (2016).
5. Touyz, R.M. & Montezano, A.C. Vascular Nox4: a multifarious NADPH oxidase. *Circ Res* **110**, 1159-1161 (2012).

6. Kuroda, J., *et al.* NADPH oxidase 4 (Nox4) is a major source of oxidative stress in the failing heart. *Proc Natl Acad Sci U S A* **107**, 15565-15570 (2010).
7. Matsushima, S., *et al.* Broad suppression of NADPH oxidase activity exacerbates ischemia/reperfusion injury through inadvertent downregulation of hypoxia-inducible factor-1alpha and upregulation of peroxisome proliferator-activated receptor-alpha. *Circ Res* **112**, 1135-1149 (2013).
8. Teixeira, G., *et al.* Therapeutic potential of NADPH oxidase 1/4 inhibitors. *Br J Pharmacol* **174**, 1647-1669 (2017).
9. Gray, S.P., *et al.* Combined NOX1/4 inhibition with GKT137831 in mice provides dose-dependent reno- and atheroprotection even in established micro- and macrovascular disease. *Diabetologia* **60**, 927-937 (2017).
10. You, Y.H., Quach, T., Saito, R., Pham, J. & Sharma, K. Metabolomics Reveals a Key Role for Fumarate in Mediating the Effects of NADPH Oxidase 4 in Diabetic Kidney Disease. *J Am Soc Nephrol* **27**, 466-481 (2016).
11. Wang, W., *et al.* Hypoxia-inducible factor 1 in clinical and experimental aortic aneurysm disease. *J Vasc Surg* **68**, 1538-1550 e1532 (2018).
12. Li, X., *et al.* Curcumin attenuates the development of thoracic aortic aneurysm by inhibiting VEGF expression and inflammation. *Mol Med Rep* **16**, 4455-4462 (2017).
13. Daugherty, A., Chen, Z., Sawada, H., Rateri, D.L. & Sheppard, M.B. Transforming Growth Factor-beta in Thoracic Aortic Aneurysms: Good, Bad, or Irrelevant? *J Am Heart Assoc* **6**(2017).
14. Cook, J.R., *et al.* Dimorphic effects of transforming growth factor-beta signaling during aortic aneurysm progression in mice suggest a combinatorial therapy for Marfan syndrome. *Arterioscler Thromb Vasc Biol* **35**, 911-917 (2015).
15. Lassegue, B., San Martin, A. & Griendling, K.K. Biochemistry, physiology, and pathophysiology of NADPH oxidases in the cardiovascular system. *Circ Res* **110**, 1364-1390 (2012).
16. Dostmann, W.R., *et al.* Highly specific, membrane-permeant peptide blockers of cGMP-dependent protein kinase Ialpha inhibit NO-induced cerebral dilation. *Proc Natl Acad Sci U S A* **97**, 14772-14777 (2000).
17. Taylor, M.S., *et al.* Inhibition of cGMP-dependent protein kinase by the cell-permeable peptide DT-2 reveals a novel mechanism of vasoregulation. *Mol Pharmacol* **65**, 1111-1119 (2004).
18. Schmittgen, T.D. & Livak, K.J. Analyzing real-time PCR data by the comparative C(T) method. *Nat Protoc* **3**, 1101-1108 (2008).
19. Prisyazhna, O., Rudyk, O. & Eaton, P. Single atom substitution in mouse protein kinase G eliminates oxidant sensing to cause hypertension. *Nat Med* **18**, 286-290 (2012).
20. Kalyanaraman, H., Zhuang, S., Pilz, R.B. & Casteel, D.E. The activity of cGMP-dependent protein kinase Ialpha is not directly regulated by oxidation-induced disulfide formation at cysteine 43. *J Biol Chem* **292**, 8262-8268 (2017).
21. Pfeifer, A., *et al.* Defective smooth muscle regulation in cGMP kinase I-deficient mice. *EMBO J* **17**, 3045-3051 (1998).
22. Chung, A.W., *et al.* Endothelial dysfunction and compromised eNOS/Akt signaling in the thoracic aorta during the progression of Marfan syndrome. *Br J Pharmacol* **150**, 1075-1083 (2007).
23. Vogt, J., Traynor, R. & Sapkota, G.P. The specificities of small molecule inhibitors of the TGF- β and BMP pathways. *Cell Signal* **23**, 1831-1842 (2011).
24. Pedruzzi, E., *et al.* NAD(P)H oxidase Nox-4 mediates 7-ketocholesterol-induced endoplasmic reticulum stress and apoptosis in human aortic smooth muscle cells. *Mol Cell Biol* **24**, 10703-10717 (2004).
25. Oller, J., *et al.* Nitric oxide mediates aortic disease in mice deficient in the metalloprotease Adamts1 and in a mouse model of Marfan syndrome. *Nat Med* **23**, 200-212 (2017).
26. Moore-Morris, T., *et al.* Resident fibroblast lineages mediate pressure overload-induced cardiac fibrosis. *J Clin Invest* **124**, 2921-2934 (2014).

REVIEWERS' COMMENTS:

Reviewer #1 (Remarks to the Author):

The authors have significantly addressed the comments I made during the initial review. I guess I would have liked to see resistance blood vessel myograph studies given the newly analysed blood pressure data observations. But, this is a relatively minor point when considering all the other data presented in the manuscript that reports and interesting observation that I think will be of interest to many in the community.

Reviewer #2 (Remarks to the Author):

N/A

Reviewer #3 (Remarks to the Author):

1. The reviewer's major concern is that the changes in aortic dimensions are very modest in both models. It should also be noted that the changes in aortic dimensions in Figure 1 are very modest and are probably similar to changes in the cardiac cycle. Although the authors' response claims that the measurements were recorded in diastole, the reviewer could not find this description in the methods section of the manuscript.
2. The authors changed "aortic degeneration" to "aortic pathology" in the text. "Aortic degeneration" in the title should also be changed.
3. Statistical analysis: in the authors' response, it was stated that if $N > 7$, normality and equal variance tests were performed, but in the Method section, it stated " $N > 18$ ". The rationale to decide for normality and equal variance tests based on specific sample size is unclear.
4. The term "showed a trend towards improving survival..." on page 9 should be deleted. If it is not statistically significant, it should be stated "no difference".

5. For C57BL mice, 6 months of age is equivalent to ~ 30 years of age in human, and for 12 month old mice, their age is equivalent to ~45 years in human. Hence, the authors should reconsider the use of the term “aging” in reference to their studies.

Re: Revision of Manuscript # NCOMMS-18-34576-B:
June 18, 2019

Response to Reviewers (in italics):

Reviewer #1 (Remarks to the Author):

The authors have significantly addressed the comments I made during the initial review. I guess I would have liked to see resistance blood vessel myograph studies given the newly analysed blood pressure data observations. But, this is a relatively minor point when considering all the other data presented in the manuscript that reports and interesting observation that I think will be of interest to many in the community.

Reviewer #2 (Remarks to the Author):

N/A

Reviewer #3 (Remarks to the Author):

1. The reviewer's major concern is that the changes in aortic dimensions are very modest in both models. It should also be noted that the changes in aortic dimensions in Figure 1 are very modest and are probably similar to changes in the cardiac cycle. Although the authors' response claims that the measurements were recorded in diastole, the reviewer could not find this description in the methods section of the manuscript.

We now state that "moderate aortic dilation became apparent at 12 months," referring to Fig. 1e,f (page 4). In the TAC model, we used four-month-old mice, which did not show aortic dilation prior to the procedure, and after the procedure dilation was the same between wild type and surviving mutant mice, but more Prkg1^{RQ/+} mice died of aortic rupture. We previously mentioned in the legend to Fig. 1 that ultrasound measurements were performed in end-diastole; we now include this statement also in the Methods Section (page 12).

2. The authors changed "aortic degeneration" to "aortic pathology" in the text. "Aortic degeneration" in the title should also be changed.

The title was changed to "Aortic Pathology from Protein Kinase G Activation Is Prevented by an Antioxidant Vitamin B₁₂ Analog" – and was shortened to 15 words including "G" and "B12".

3. Statistical analysis: in the authors' response, it was stated that if $N > 7$, normality and equal variance tests were performed, but in the Method section, it stated " $N > 18$ ". The rationale to decide for normality and equal variance tests based on specific sample size is unclear.

We apologize for the typo – the data were tested for normality using the D'Agostino-Pearson omnibus normality test when appropriate for $n > 7$, since the D'Agostino test requires eight or more values.

4. The term "showed a trend towards improving survival..." on page 9 should be deleted. If it is not statistically significant, it should be stated "no difference".

We deleted the statement suggesting a trend towards improving survival and now state "the survival of NAC-treated mutant mice subjected to TAC did not differ from that of untreated mice" (page 8).

5. For C57BL mice, 6 months of age is equivalent to ~ 30 years of age in human, and for 12 month old mice, their age is equivalent to ~45 years in human. Hence, the authors should reconsider the use of the term "aging" in reference to their studies.

We now state "middle-aged Prkg1^{RQ/+} mice recapitulated aortic changes observed in patients heterozygous for the PKG^{RQ} mutation (page 4).